# Intelligent perceptual textiles based on ionic-conductive and strong silk fibers

Haojie Lu[1], Yong Zhang[1], Mengjia Zhu[1], Shuo Li[1], Huarun Liang[1], Peng Bi[1], Shuai Wang[1], Haomin Wang[1], Linli Gan[1], Xun-En Wu[1] & Yingying Zhang [1] ✉

Endowing textiles with perceptual function, similar to human skin, is crucial for the development of next-generation smart wearables. To date, the creation of perceptual textiles capable of sensing potential dangers and accurately pinpointing finger touch remains elusive. In this study, we present the design and fabrication of intelligent perceptual textiles capable of electrically responding to external dangers and precisely detecting human touch, based on conductive silk fibroin-based ionic hydrogel (SIH) fibers. These fibers possess excellent fracture strength (55 MPa), extensibility (530%), stable and good conductivity (0.45 S·m⁻¹) due to oriented structures and ionic incorporation. We fabricated SIH fiber-based protective textiles that can respond to fire, water, and sharp objects, protecting robots from potential injuries. Additionally, we designed perceptual textiles that can specifically pinpoint finger touch, serving as convenient human-machine interfaces. Our work sheds new light on the design of next-generation smart wearables and the reshaping of human-machine interfaces.

Perception is a crucial function of human skin. While flexible film-based electronics have been developed as electronic skins[1–6], textile-based electronics can provide superior flexibility, air permeability, and comfort for wearable electronics[7–10]. Integrating perceptual function into textiles will revolutionize the way humans interact with electronic devices and contribute to the further development of smart wearables. Recent advances in electronic textiles have enabled communication, sensing, display, power supply, and other functions[11–14]. However, perceptual textiles capable of exclusively and accurately pinpointing human touch and helping humans/robots recognize and respond to dangers, such as fire, water, and fracture, have yet to be reported.

To address this challenge, flexible, strong, and conductive fibers are required as the basic units. While metallic wires and nanocarbon-based fibers offer good conductivity, they are prone to electrical and mechanical failures under tension or cyclic bending deformation[15–17]. Geometric designs, such as serpentine, kirigami, or wrinkled structures, can impart certain stretchability to fibers[18–21], but pose challenges in terms of compatibility with textiles and increase the risk of failure and cost. Alternatively, ion gels feature inherent flexibility,

conductivity, transparency, and robustness[22,23]. They can transfer electricity through mobile ions, similar to biological tissues[24,25]. Therefore, robust gel fibers that are biocompatible, mechanically strong, and highly ionic-conductive have the potential to be woven into the desired intelligent textiles.

In this work, we prepared a highly strong, conductive, and stable silk fibroin-based ionic hydrogel (SIH) fiber and realized the fabrication of intelligent perceptual textiles capable of precisely detecting external dangers and human touch. The SIH fiber, composed of natural silk fibroin, ionic liquid ([Emim]BF₄), and glycerol, was prepared through a continuous wet spinning process. It has a semi-crystalline and oriented structure similar to natural silkworm silk, leading to a high tensile strength of 4 MPa, which can be further increased to 55 MPa via post-stretching. The SIH fiber also possesses a notable extensibility of up to 530%, more than 20 times that of natural silkworm silk (~25%), which can be ascribed to the plasticization of ionic liquid [Emim]BF₄, glycerol, and water. Importantly, the incorporation of [Emim]BF₄ endows the SIH fibers with stable and high ionic conductivity up to 0.45 S·m⁻¹. Furthermore, we demonstrated the application of SIH fibers in

[1]Key Laboratory of Organic Optoelectronics and Molecular Engineering of the Ministry of Education, Department of Chemistry, Tsinghua University, Beijing, P. R. China. ✉e-mail: yingyingzhang@tsinghua.edu.cn

perceptual textiles. The circuits with integrated SIH fibers can show instantaneous and characteristic responses to stimuli such as fire burning, water immersing, sharp object cutting, and finger touching, endowing the SIH fiber-based intelligent textiles with sensing capabilities for protection. Besides, we demonstrated that the designed perceptual textiles are capable of precisely and specifically detecting the occurrence and location of touch. Therefore, the successful fabrication of silk fibers with intrinsic ion conductivity and excellent mechanical properties will promote a significant advancement in the functionalization and utilization of silkworm silk fibers. Furthermore, the development of perceptual textiles capable of exclusively and accurately pinpointing human touch will revolutionize human-machine interfaces, offering excellent flexibility and comfort, bringing great convenience in intelligent living to humans.

## Results

### Preparation of strong and conductive SIH fibers

To create perceptual textiles, we first fabricated flexible, strong, and conductive SIH fibers (Fig. 1a). Nature silk fibroin fibers possess remarkable mechanical performance due to their oriented semi-crystalline and hierarchical structures[26,27]. To make SIH fibers, we extracted silk fibroin from silkworm cocoons and prepared a fibroin/formic acid solution as the spinning dope. The SIH fibers was prepared using a continuous wet spinning method. The spinning dope was extruded continuously into a $CH_3OH$ coagulation bath to form a regenerated silk fibroin fiber. This fiber was then passed through an ionic liquid bath containing 1-ethyl-3-methylimidazolium tetrafluoroborate ([Emim]$BF_4$), glycerol, and water for solvent exchange. The resulting SIH fiber was collected continuously. Figure 1b shows a reel of the obtained SIH fiber. We varied the concentration of the ionic liquid bath to optimize the properties of the fibers, which are donated as X-S, where X is the molar concentration of [Emim]$BF_4$ in the bath.

The as-prepared SIH fibers have a round cross-section, a smooth surface, good transparency, excellent flexibility, outstanding weavability, and remarkable ionic conductivity. The scanning electron microscopy (SEM) images (Fig. 1c, d and Supplementary Fig. 1) reveal that the SIH fibers have circular cross-sections and smooth surfaces. The diameter of

SIH fibers increases slightly when the concentration of [Emim]$BF_4$ increases (Supplementary Fig. 2a). Besides, the diameters of all SIH fibers are greater than that of bare regenerated silk fibers not subjected to solvent exchange within the ionic liquid reservoir (Supplementary Fig. 2 and 3), indicating that the treatment of fibers within ionic liquid solution improved their diameters. In addition, we can tune the fiber diameter by adjusting the spinneret diameters and feeding rates (Supplementary Fig. 4). We also quantified the weight variation during the solvent exchange process within the ionic liquid reservoir (Supplementary Fig. 5). The data elucidates a progressive trend in unit-length weight as the ionic liquid concentration increases. Subsequently, according to the composition of ionic liquid solution (Supplementary Fig. 6 and Supplementary Table 1) and the content of fibroin and solution within different SIH fibers (Supplementary Fig. 5), the detailed composition of SIH fibers was calculated and the data was shown in Supplementary Fig. 7 and Supplementary Table 2. An escalation in the concentration of ionic liquid yields an increment in the ionic liquid content within SIH fibers. Concomitantly, the ratios of silk fibroin, glycerol, and water within these fibers exhibit a diminishing trend as the ionic liquid content amplifies. Figure 1e is a photograph of a SIH fiber coil, demonstrating the fiber's excellent transparency and flexibility. Figure 1f shows a $9 \times 9\,cm^2$ plain weave textile made of SIH fibers, demonstrating their remarkable weavability. In addition to good transparency and flexibility, the SIH fibers also possess excellent ionic conductivity. Figure 1g shows an image of "SILK" patterned display devices (Supplementary Movie 1) with SIH fibers serving as the external electrodes.

### Composition and structure of SIH fibers

We conducted a comprehensive investigation into the composition and structure of SIH fibers prepared under various ion exchange conditions. The [Emim]$BF_4$ was successfully incorporated into SIH fibers, which was confirmed by thermogravimetry (TG) and energy dispersive spectrometry (EDS). Figure 2a, b shows the TG and differential thermogravimetry (DTG) curves of different SIH fibers, which show distinct stages of weight loss corresponding to the varying stability of the components in the fibers. Specifically, weight loss at 30 °C, 180 °C, and 290 °C can be ascribed to free water, fibroin, and glycerol,

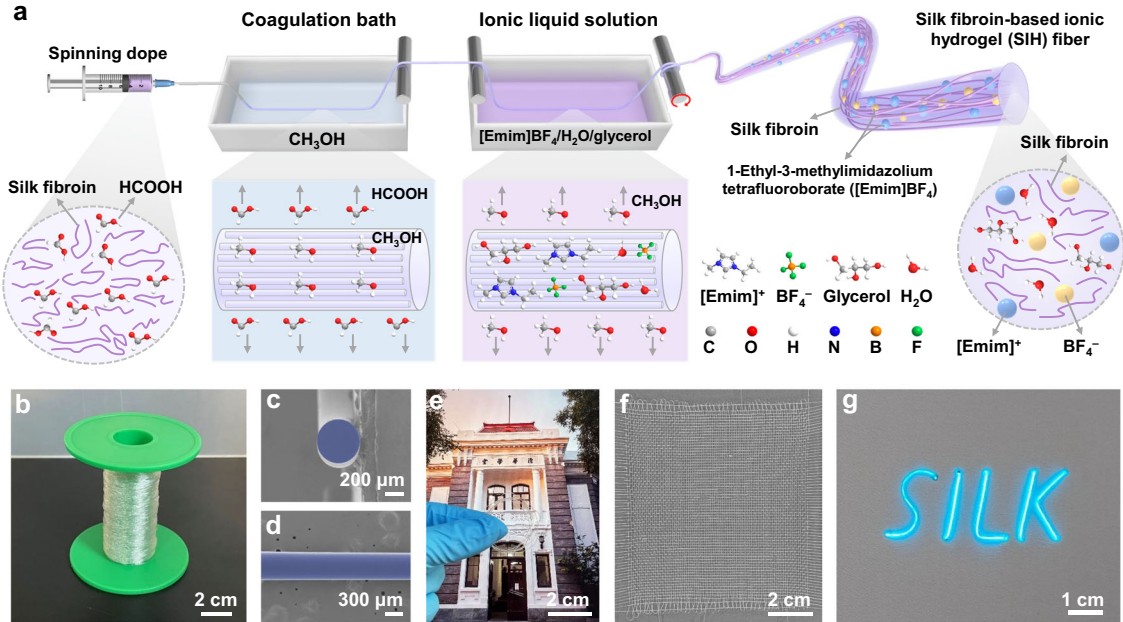

**Fig. 1 | Fabrication of silk fibroin-based ionic hydrogel (SIH) fibers. a** Scheme diagram for the fabrication of SIH fibers. **b** Photograph of a roll of SIH fibers. **c, d** Typical SEM images of SIH fibers, showing the cross-sectional (**c**) and surface (**d**) morphology. **e** Photograph of a SIH fiber coil on a color image, showing its transparency and appearance. **f** A plain weave textile made of SIH fibers, demonstrating the weavability of SIH fibers. **g** An electroluminescent "SILK" pattern with SIH fibers serving as flexible and transparent electrodes.

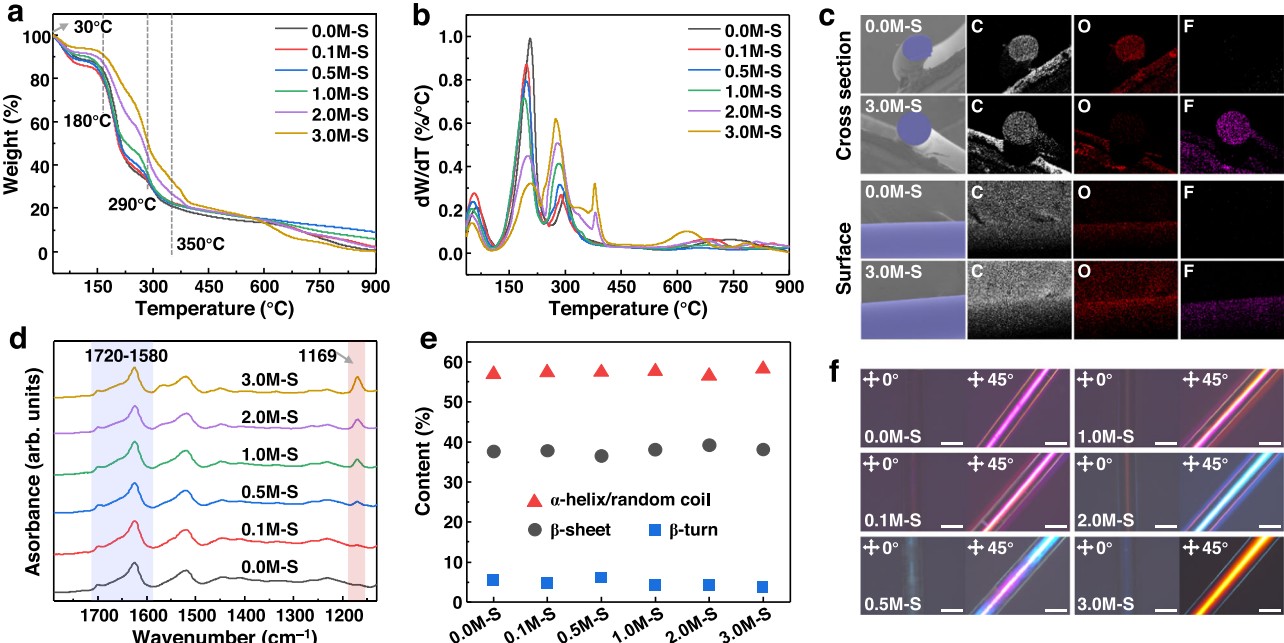

**Fig. 2 | Composition and structure analysis of different SIH fiber samples. a, b** TG (**a**) and DTG (**b**) curves. **c** SEM images and corresponding EDS mapping. **d** FTIR spectra. **e** Secondary structure contents of different SIH fiber samples. **f** POM images of SIH fibers. Scale bar, 500 μm.

respectively[28]. Particularly, weight loss at 350 °C is attributed to the [Emim]BF$_4$[29]. Furthermore, the 3.0M-S exhibited a lower weight loss than the controlled fibers (0.0M-S) in the temperature range of 30 °C – 375 °C, indicating improved thermal stability of SIH fibers upon incorporating [Emim]BF$_4$. EDS analysis also confirmed the presence of [Emim]BF$_4$ in the SIH fibers. As depicted in Fig. 2c, both 0.0M-S and 3.0M-S contained C and O as the main elements in fibroin. However, the element F was only present in 3.0M-S, indicating the successful incorporation of [Emim]BF$_4$. Additionally, EDS images revealed uniform distribution of [Emim]BF$_4$ in the fibers, from sections to surfaces. X-ray photoelectron spectroscopy (XPS) tests were also performed to analyze the distribution of [Emim]BF$_4$ in the fibers. The XPS curves recorded from the surface to the inside of 3.0M-S showed characteristic peaks of elements F 1$s$ and B 1$s$ (Supplementary Fig. 8 and 9), indicating the successful incorporation of [Emim]BF$_4$. In contrast, no B 1$s$ and F 1$s$ signals were observed in the XPS curves of 0.0M-S (Supplementary Fig. 10 and 11).

Figure 2d shows the Fourier transform infrared spectroscopy (FTIR) spectra of different SIH fiber samples. All the samples exhibit a peak at 1720 – 1580 cm$^{-1}$, which is ascribed to the amide I band of silk fibroin[30,31]. The peak at 1169 cm$^{-1}$, which only appears in the FTIR spectra of SIH fibers treated with ionic liquid, is related to the asymmetric vibrations of C – N – C in the rings of [Emim]$^+$[32]. The intensity of this peak increases with a higher concentration of [Emim]BF$_4$, providing further evidence for the successful incorporation of the ionic liquid in the silk fibroin hydrogel fibers.

The SIH fibers possess a semi-crystalline and oriented structure, as revealed by deconvolution of the FTIR amide I band and polarizing optical microscopy (POM) tests. Deconvolution of the FTIR amide I band (Supplementary Fig. 12) shows similar contents of crystalline (β-sheet) and amorphous (α-helical/random coil) structures in all the samples (Fig. 2e), indicating that the incorporation of [Emim]BF$_4$ did not have a significant influence on the crystallinity of the fibers. POM images of the fibers (Fig. 2f) show that the biaxial crystals in the fibers are preferentially aligned along the fiber axis rather than in other directions. When the axis of the fibers was parallel to the polarizer, the fibers could not be observed in the POM image, but appear after a 45° rotation, confirming the oriented structure of the fibers.

## Mechanical and electrical properties of SIH fibers

SIH fibers exhibit exceptional mechanical properties, characterized by high tensile strength and remarkable extensibility. Figure 3a depicts the typical strain-stress curves of various SIH fiber samples. The tensile strength and elongation at break of 0.0M-S are 2.5 MPa and 450%, respectively. Notably, with an increase in the concentration of [Emim]BF$_4$ to 0.1 M, the tensile strength and elongation at break increase to 4 MPa and 530%. Further, an increase in [Emim]BF$_4$ concentration results in a minor reduction in mechanical properties. These findings demonstrate that the prepared hydrogel silk fibers possess superior stretchability compared to natural silk fibroin fibers[33]. We further studied the influence of ionic liquid concentration on the mechanical properties of the SIH fibers. With the introduction of a small amount of ionic liquid (0.1 M), the content of glycerol and water to silk fibroin within the fiber notably escalates (Supplementary Fig. 7). However, as the concentration of ionic liquid increases further, both the content of glycerol and water decrease. Glycerol and water act as plasticizers within the fiber, mitigate crack propagation along the radial axis during stretching, thus reducing the brittleness of fibers. Subsequent elevation of the ionic liquid concentration decreased the content ratios of glycerol, water, and silk fibroin, and increased the diameter of silk fibers, leading to a progressive deterioration in the mechanical properties of fibers. Consequently, the mechanical performance of 0.1M-S fibers are better than other fibers. Besides, the SIH fibers showed obvious hysteresis during cyclic loading and unloading of tensile stress (Supplementary Fig. 13). The stress-strain curves show that little plastic deformation occurred in the SIH fibers after the first training cycle, indicating the good fatigue resistance.

Additionally, the mechanical strength of SIH fibers can be further enhanced by a stretching treatment before ion exchange. Our previous study has proved that stretching can improve the crystalline orientation in the fibers, thus improving their strength[34]. In this study, we subjected regenerated silk fibers to stretching with different ratios prior to ion exchange. The diameters of fibers gradually decrease as the stretching ratio increases from 0% to 200% (Supplementary Fig. 14). The silk fibers in all groups possess uniform diameters and smooth surfaces (Supplementary Fig. 15). Figure 3b shows the strain-stress curves of SIH fibers with varied stretching ratios. Stretching the

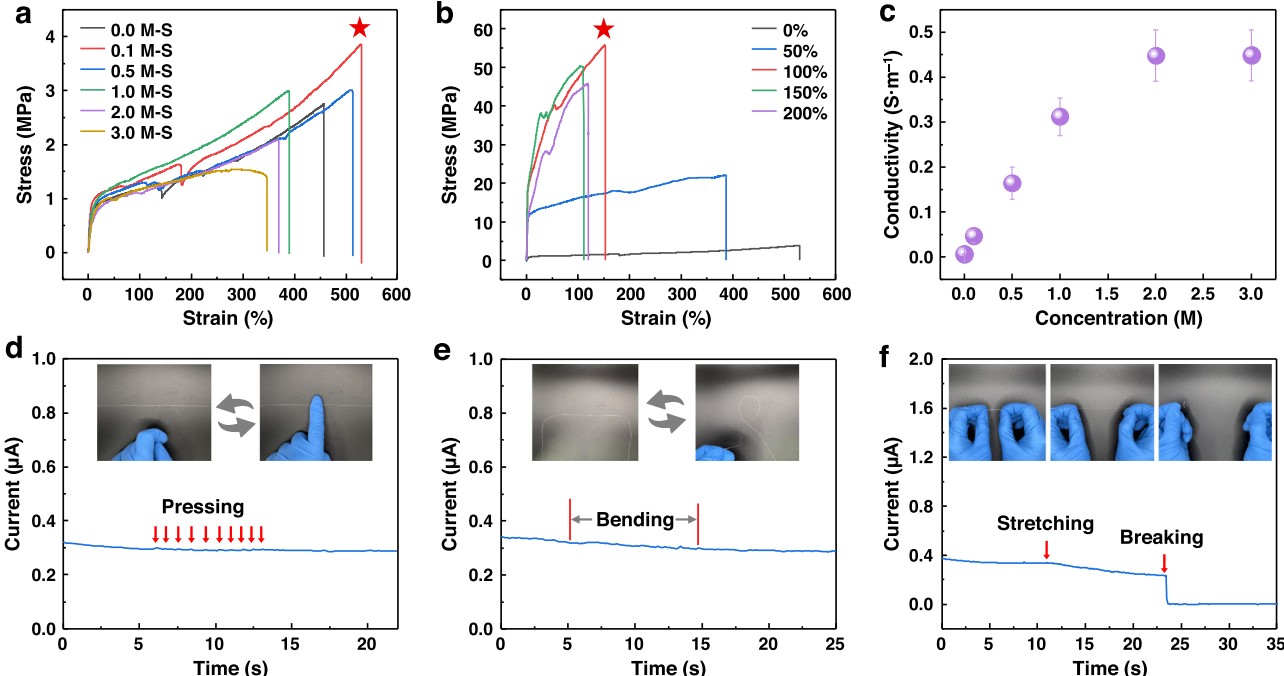

**Fig. 3 | Mechanical and electrical properties of SIH fibers. a** Strain-stress curves of SIH fibers with various concentrations of [Emim]BF$_4$. **b** Strain-stress curves of SIH fibers treated with different stretching ratios during the preparation process. **c** Conductivity versus [Emim]BF$_4$ concentration plots of SIH fibers. The error bars show the standard deviation ($n = 3$ for each group). **d–f** Evolution of the current when the SIH fiber is pressed (**d**), bent (**e**), stretched (**f**), showing the stable conductivity under mechanical deformations. The insets show the corresponding images.

fibers by 100% leads to the highest tensile strength of 55 MPa, which is several times higher than previously reported hydrogel fibers (less than 10 MPa)[35–40]. However, excessive stretching (stretching ratio ≥ 150%) decreased both strength and elongation, likely due to the introduction of defects. Despite this, all stretched SIH fibers demonstrate significantly higher strength than their non-stretched counterparts.

Moreover, the SIH fibers exhibited excellent ionic conductivity. As shown in Fig. 3c, from 0.0M-S to 1.0 M-S, the conductivity progressively increases, which is attributed to the rapid increase in the ionic liquid content of the SIH fibers (Supplementary Fig. 7). As the concentration reaches 2.0 M, the rate of increase in conductivity of the SIH fibers decreases, which is ascribed to that the enhanced interaction between oppositely charged ions in high concentration of ionic liquid solutions hampers ion mobility. The electrical conductivity is up to 0.45 S·m$^{-1}$ for 2.0M-S, comparable to previously reported conductive hydrogels[25,38,41].

Remarkably, the conductivity of SIH fibers is highly stable, which is crucial for practical applications. We investigated the effect of long-term storage on the ionic conductivity of SIH fibers and found a slight decrease from 0.45 S·m$^{-1}$ to 0.43 S·m$^{-1}$ after being stored at 43% humidity for 21 days (Supplementary Fig. 16). It is noted that 43% is a relatively low humidity value in actual environment (Supplementary Fig. 17 and 18). The excellent stability of the fiber can be ascribed to the existence of water retaining glycerol. Besides, the water-soluble ionic liquids and the silk fibroin that contains lots of amino acid residues (~30%) with hydrophilic groups also reduce the evaporation of water. Moreover, the electrical property of the fibers is insensitive to various deformations as demonstrated by their stable current to pressing or bending (Fig. 3d, e). Stretching leads to a slight decrease in current, likely due to the increased length and decreased diameter, but the fiber is still able to maintain a moderate current until it broke (Fig. 3f). These observations prove the high reliability and robustness of the SIH fibers for practical applications.

Performance comparison proves that our fibers have excellent mechanical and electrical properties (Supplementary Fig. 19 and Supplementary Table 3). In contrast to various other ionic conductive materials, our fibers exhibit higher strength and moderate elongation at break, effectively addressing the existing gaps in the mechanical properties within the related domain (Supplementary Fig. 19a). Besides, our fiber with stretching boasts an absolute mechanical strength advantage over a spectrum of ionic conductive materials (Supplementary Fig. 19b). Supplementary Fig. 19c underscores that our fibers combine heightened strength with a moderate level of conductivity, effectively bridging the existing gaps in the relevant domain. With exceptional mechanical and electrical properties, our fibers are poised to play an important role in the realm of smart textiles.

## Perceptual textiles for recognizing dangers
Furthery, we explored the potential applications of SIH fibers in the field of intelligent textiles. These fibers possess the unique ability to respond to various hazardous situations, such as fire, water, and sharp objects. As a result, they can be used to fabricate smart textiles that protect humans and robots from burns, soaking, and skin scratches. For a proof of concept, we fabricated an intelligent protective glove for a robot hand, which incorporated SIH fibers (Fig. 4a) and integrated a circuit system (Supplementary Fig. 20). When exposed to hazardous conditions, the intelligent glove generates corresponding electrical signals. As shown in Fig. 4b, the current of SIH fiber increases suddenly when in contact with fire and then drops to zero. Figure 4c depicts the structure evolution of the SIH fiber when being exposed to fire. The temperature of the fire is 981 °C. According to Fig. 2a, the SIH fibers were decomposed and carbonized at this temperature. Therefore, the rise of current in Fig. 4b is attributed to the rise of temperature and the carbonization of SIH fiber. The decrease in current to zero is ascribed to the failure of SIH fiber. Water can also pose a danger to robots by causing a short circuit. We observe that when a drop of water falls on the SIH fiber, the current increases instantaneously and then gradually. As shown in Fig. 4d, three obvious steps were observed when three drops of water dripped in sequence. The reason for this is that the SIH

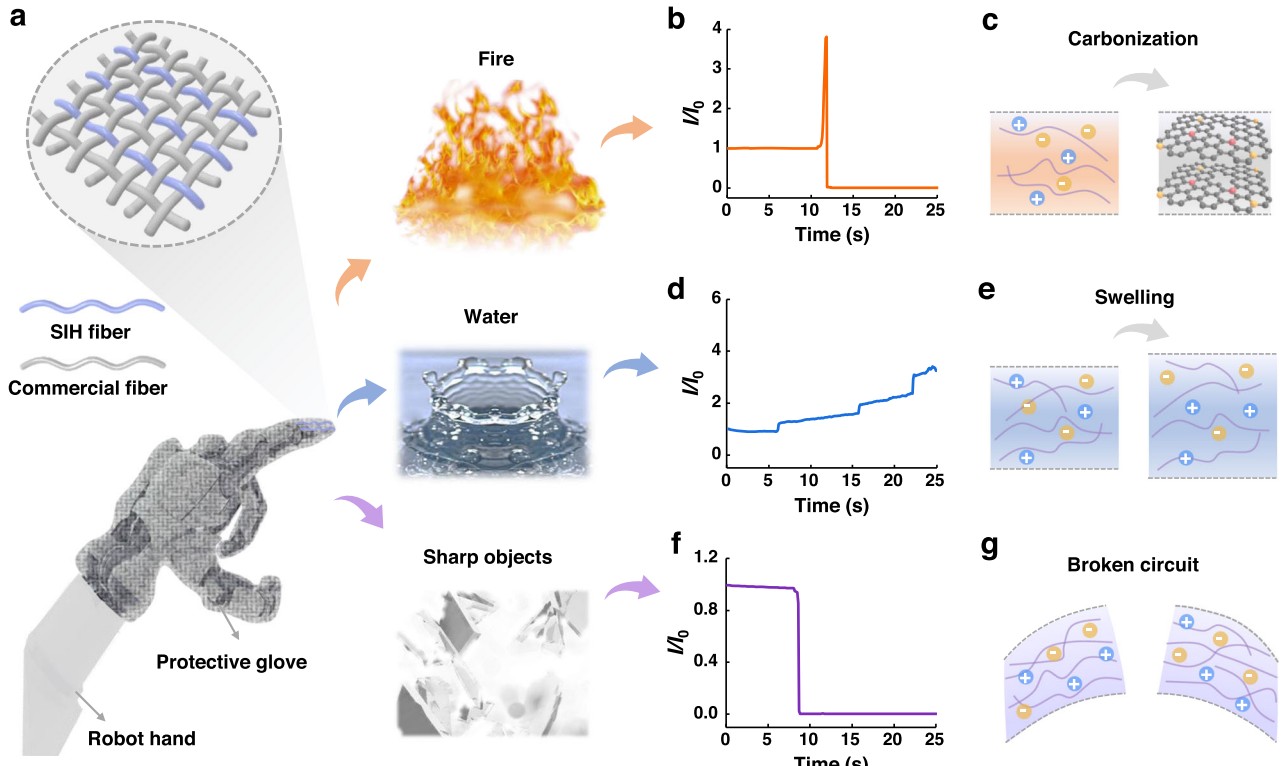

**Fig. 4 | Electrical responses of SIH fibers to fire, water, and cut, showing potential applications in recognizing and warning of dangers. a** Illustration shows a robot hand wearing a piece of SIH fiber-based textile which can detect fire burning, water immersing, and sharp object cutting. **b–g** Characteristic electrical responses of SIH fibers to fire (**b, c**), water (**d, e**), and cut (**f, g**) and the corresponding working mechanism.

fiber swelled after absorbing water, improving their ion transfer capacity (Fig. 4e). The current can be determined by Eq. 1.

$$I = \frac{\pi D^2 U \sigma}{4 \times 10^{10} \times L} \quad (1)$$

Where $D$ (μm) and $L$ (cm) are the diameter and length of the fiber, respectively. $U$ (V) is the applied voltage on the fiber (1 V), $I$ (A) and σ are the output current and conductivity of the fiber. The addition of water droplets diminishes the ionic liquid concentration, thus reducing σ. $I$ concurrently increased, which indicates an augmentation in $D$. That is, the amplification of current by the increased area far outweighs the diminishing effect of conductivity decrease on current, leading to a net increase in the output current of the fiber. Besides, sharp objects can break through the robot's skin, which can also be detected by the SIH fiber-integrated glove. As shown in Fig. 4f, the current drops to zero when the SIH fiber breaks, indicating the breakage of the conductive paths (Fig. 4g). According to the different responses of the current to various stimuli, the SIH fiber-integrated textile can detect and differentiate between different dangerous situations.

## Perceptual textiles for human-machine interfaces

We further designed and fabricated perceptual textiles using SIH fibers, which are capable of accurately and specifically detecting the occurrence and location of human touch, making them ideal for wearable human-machine interfaces. As depicted in Fig. 5a, a man is able to control a robot hand by touching these perceptual textiles. The perceptual textiles can be produced by integrating a SIH fiber into a commercial fabric or weaving SIH fibers into a textile. Figure 5b shows the equivalent circuit for a perceptual textile based on a single SIH fiber. The SIH fiber was connected to the circuit at both ends, and an alternating voltage was applied. When the SIH fiber-based textiles are

worn on the human body, the ground electrode of the power is connected to the skin. A closed circuit is formed when a finger touches the fiber. The SIH fiber is divided into two parts by the touch point. The location of the touch point ($x$) can be determined using the Eqs. 2 and 3.

$$k = \frac{U_1}{U_2} = \frac{I_1 R_0}{I_2 R_0} = \frac{U/(R_0 + xR)}{U/[R_0 + (1-x)R]} = \frac{R_0 + (1-x)R}{R_0 + xR} \quad (2)$$

$$x = \frac{R_0 + R - kR_0}{(k+1)R} \quad (3)$$

Where $U_1$ and $U_2$ represent the voltages of the left and right fixed resistors, $U$ represents the alternating voltage of power, $I_1$ and $I_2$ represent the currents of the left and right fixed resistors, and $R_0$ and $R$ represent the resistances of the fixed resistor and SIH fiber. By sequentially touching points of $X$ at 0.1 intervals from 0.1 to 0.9, voltage signals can be induced and recorded. Accordingly, the values of $k$ and $x$ can be calculated. As shown in Fig. 5c, the calculated values of $x$ are in agreement with the theoretical values for $X$, indicating that the perceptual textile based on a single SIH fiber can precisely detect the location of touch.

Furthermore, we designed and successfully produced a perceptual textile by weaving SIH fibers into a plain-woven fabric. Figure 5d shows the equivalent circuit of the perceptual textile under finger touch. When a finger touches the textile at point ($a$, $b$), the voltage signals ($U_1$, $U_2$, $U_3$, and $U_4$) are recorded. The values of $a$ and $b$ are positively correlated with $(U_1 + U_3)/(U_2 + U_4)$ and $(U_1 + U_2)/(U_3 + U_4)$, respectively. Therefore, according to the values of $(U_1 + U_3)/(U_2 + U_4)$ and $(U_1 + U_2)/(U_3 + U_4)$ induced by touch, the touch point can be located. For a proof of concept, we divided the whole textile into nine areas that can represent different commands. As shown in Fig. 5e, the

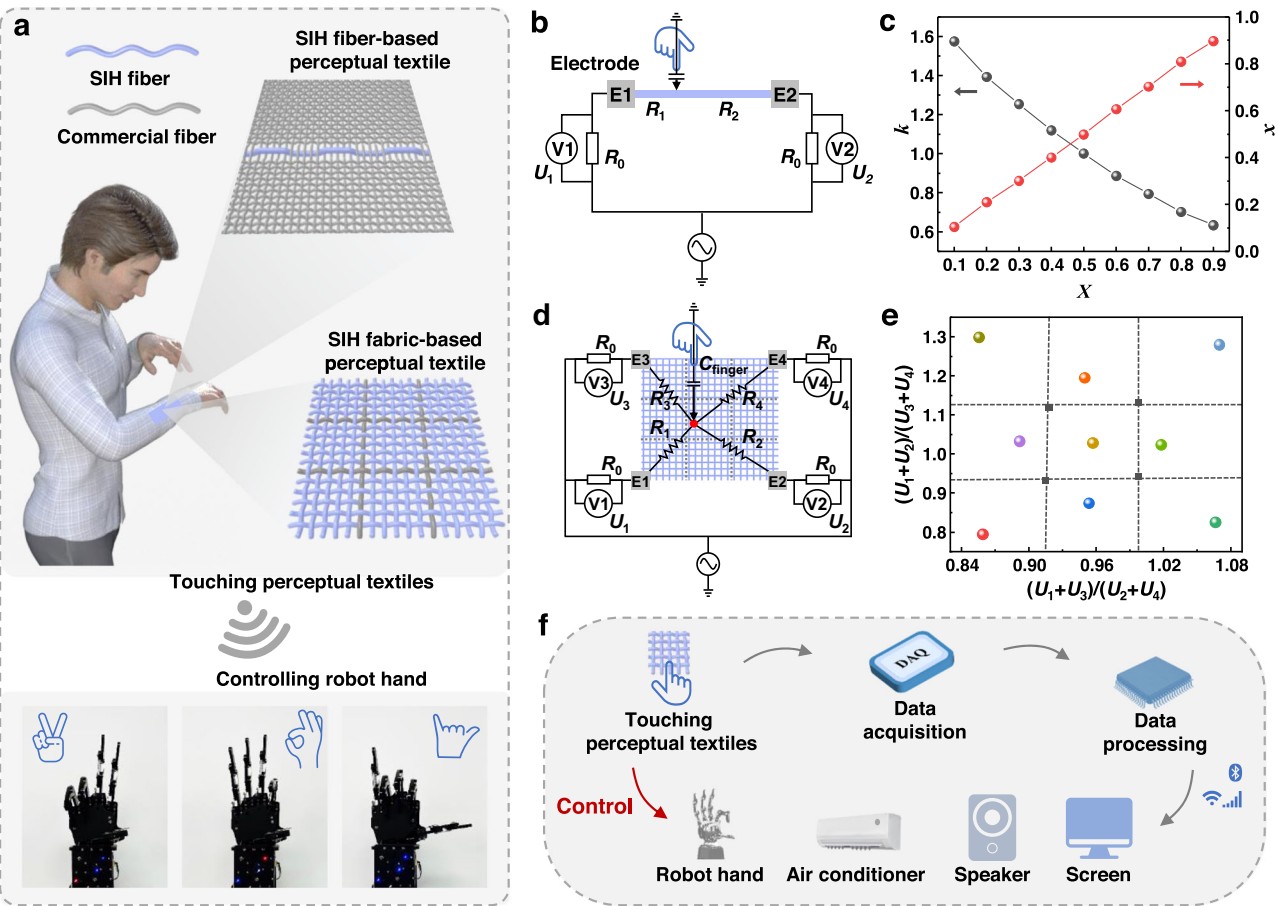

**Fig. 5 | SIH fiber-based perceptual textiles for human-machine interfaces.**
**a** Illustration showing a man touching the SIH fiber-based fibers/textiles to control the gestures of a robot hand. **b** Electrical circuit diagrams of the SIH fiber-based touch fiber. **c** The calculated scale factor $k$ and location $x$ measured according to the location of the touch point in (**b**). **d** Electrical circuit diagrams of the SIH fiber-based touch textile. **e** The calculated touch point measured according touch location in (**d**). **f** Diagram of controlling various devices by touching a SIH fiber-based perceptual textile.

location of touch points can be accurately recognized for each of the touch areas.

On the basis of exceptional perceptual performance, the SIH fiber-based perceptual textiles can serve as a convenient human-machine interface, allowing people to control various devices. As shown in Fig. 5f, the electrical signals generated by touching the SIH fiber-based textiles are recorded by a data acquisition device and processed by software in a computer or smartphone to calculate the touch point. The touch points are then associated with different commands, which are transmitted to control the terminals. For proof of concept, we successfully controlled a robot hand to perform various gestures using a SIH fiber-based textile (Supplementary Movie 2). We firmly believe, the intelligent perceptual textiles, as a portable and comfortable human-machine interface, hold immense potential for controlling intelligent wearables, smart home equipment, as well as aiding the elderly or patients for on-time assistance or medical care, promising its applications in the future.

Compared to previous tactile textiles[42], our perceptual textiles show distinctive differentiations. Firstly, the fibers comprising our textiles are stretchable ionic conductive fibers rather than stiff iron wires, endowing our textiles with inherent stretchability, flexibility, and electrical stability. Secondly, we realize the perceptual function by developing a surface capacitive sensing system instead of the piezo-resistive, piezoelectric, or triboelectric principle, which endows our textiles selectively respond only to human touch, not to any external pressure stimulus. Lastly, our perceptual textiles minimize wiring requirements and eliminate the need for complex external circuits.

## Discussion

In summary, we have developed a revolutionary perceptual textile capable of sensing dangers and human touch, utilizing a SIH fiber with stable ionic conductivity, high strength, excellent extensibility, and good flexibility. The remarkable fiber was prepared through a continuous wet spinning process, and comprised of natural derived fibroin, ionic liquid, and glycerol. Due to its semi-crystalline and oriented structure, the fiber shows an unparalleled strength of up to 55 MPa, surpassing that of other reported ionic hydrogel fibers (less than 10 MPa). The hydrogelation enables the fiber to elongate up to 530%, which is 20 times higher than that of nature silk fibers. The incorporated [Emim]BF$_4$ endows the fiber with excellent and stable conductivity of 0.45 S·m$^{-1}$. Based on the superior mechanical properties and conductivity of the SIH fiber, we have designed and fabricated perceptual textiles. The textiles can help humans or robots electrically respond to potential hazards, such as fire, water, and sharp objects. Besides, the perceptual textiles integrated a surface-capacitive touch circuit system can detect and locate finger touch, making them ideal for use in wearable human-machine interfaces. Our perceptual textiles have been successfully used to remotely control a robot hand, showcasing their potential in applications such as remote control and communication. The perceptual textiles developed in this work represent a new and comfortable human-machine interface. Given the impressive mechanical properties, stable and good conductivity of the SIH fiber, along with the sustainability, biosafety, and biodegradability of its precursor materials, we anticipate that it will serve as an excellent candidate material for the fabrication of other functional units and

thus contribute to development of human-friendly, comfortable and all-fiber-based integrated intelligent wearable systems.

## Methods

### Materials
[Emim]BF$_4$ (98%) and LiBr (99%) were obtained from Shanghai Meryer Chemical Technology Co., Ltd. NaHCO$_3$ (≥99.8%), Cu:ZnS, formic acid (98%), glycerol (≥99%), and polyurethane (32% ± 5%) were purchased from Shanghai Aladdin Biochemical Technology Co., Ltd., Shanghai Keyan Photoelectric Technology Co., Ltd., Shanghai Macklin Biochemical Technology Co., Ltd, Shanghai Titan Scientific Co., Ltd., and Shenzhen Jitian Chemical Co., Ltd., respectively. The cocoons were obtained from Alibaba. All reagents were used directly without further purification. Deionized water was produced by Master-R and used throughout the study.

### Extraction of silk fibroin
Silk fibroin was extracted from commercial silkworm cocoons using a previously reported method[43]. In brief, the cocoons were boiled twice in a NaHCO$_3$ aqueous solution (0.5% w/w) for 30 min each time to fully remove sericin. The resulting fibroin fibers were then dried at 60 °C for 12 h. Subsequently, the fibroin fibers were dissolved in a 9.3 M LiBr solution at 80 °C for 2 h with a weight ratio of 1:10. A three-day dialysis process was carried out to achieve an ionic-free aqueous silk fibroin solution, which was then centrifuged at a speed of 6000 r·min$^{-1}$ for five minutes to remove any impurities. The supernatant was thoroughly frozen at −20 °C and placed in a freeze dryer for 48 h to remove ice and obtain fibroin foam.

### Fabrication of SIH fibers
The fibroin foam was dissolved in formic acid with a weight ratio of 12.3%. The resulting solution was mixed at a speed of 3000 r·min$^{-1}$ for 10 min, using a SpeedMixer (FlackTek, DAC 330-100 SE) instrument, to ensure thorough dissolution of the fibroin foam. The fibroin solution was used as the spinning dope, which was then extruded into a methanol coagulation bath at a speed of 300 μL·min$^{-1}$ to produce regenerated fibroin fiber. Subsequently, the fiber was immersed in water and glycerol mixed solution (volume ratio = 1:1) containing [Emim]BF$_4$. Finally, the SIH fiber was collected onto reels. The time duration of immersing fibers in the ionic liquid solution is 30 min. In order to investigate the factors influencing the diameters of fibers, we adjusted the spinneret diameters and feeding rates (Supplementary Fig. 4).

### Preparation of the electroluminescent "SILK" pattern
The prepared electroluminescent fibers have a core-sheath structure with iron wire, Cu:ZnS, and SIH fiber as the internal electrode, the intermediate electroluminescent layer, and the external electrode, respectively (Supplementary Fig. 21). ZnS:Cu powder was dispersed in aqueous polyurethane in a 1:1 weight ratio. The resulting dispersion was then mixed at 3000 r·min$^{-1}$ for 10 min. An iron wire was then immersed in the dispersion for 30 s and then dried at 60 °C for 15 min, with this process being repeated twice. The wire was then wrapped by SIH fibers which serve as the external transparent electrodes. The obtained fibers were designed in a "SILK" pattern and fixed onto a silk fabric. An alternating voltage of 200 V was used as a power source.

### Preparation and application demonstration of SIH fiber-based textiles
The SIH fiber-based protective textiles for a robot to distinguish dangers was achieved by sewing a SIH fiber into a commercial glove. The SIH fiber-based perceptual textile for humans was created by sewing a SIH fiber into a piece of commercial textile. The SIH fabric-based perceptual textile was prepared on a homemade holder in a plain weave structure by hand. In order to control the movement of the robot hand with the SIH fiber-based perceptual textile, a system was designed as shown in Fig. 5f. When a finger touches the SIH fiber-based perceptual textile, two voltages are collected by a data acquisition board. The data are processed to determine the location ($x$), which was then paired with a specific command for controlling the robot to display various gestures.

### Characterization
The cross-section and surface morphology of fibers were characterized using an SEM (ZEISS, Merlin Compact). Additionally, the same device was utilized to obtain EDS images to display the distribution of elements in the fibers. Pseudo-coloring was used for the SEM images. Surface analysis and depth profiling of fibers were performed using an XPS instrument (Ulvac-Phi Inc., PHI Quantera II). Thermal stability was evaluated by recording TG curves of the fibers, which were accomplished using a TGA/DSC simultaneous thermal analyzer (Mettler-Toledo, TGA/DSC1/1600LF). The temperature range was 30 – 900 °C, and the heating rate was 10 °C·min$^{-1}$ in air. FTIR spectra was collected using an instrument from PerkinElmer, specifically the Spotlight 400 model. The strain-load curves were recorded by a mechanical testing machine (Shimadzu, AGS-X). Optical microscopy (Leica, DM2500M) was employed to observe the diameters of fibers and record POM images to indicate the alignment of fibers. The electrical conductivity and stability tests were performed on an electrochemical workstation (CHI760E). The samples utilized for testing conductivity were the SIH fibers without being stretched. During the electrical performance evaluation, the gauge length of SIH fibers is 20 mm, with their ends connected to conductive wires through the conductive silver paste. The $i$-$t$ curves of various SIH fibers were measured. The testing procedure entailed an initial voltage of 1 V, a sample interval of 0.1 s, and a run time of 30 s.

## Data availability
The data of morphology, diameters, compositions, FTIR deconvolution, XPS analyses, cyclic tensile tests, storge stability, city humidity, and performance comparison are provided in the Supplementary Information. Additional data are available from the corresponding author upon request.

## Code availability
The codes for human-machine interaction used in this study are available from the corresponding authors.

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

## Acknowledgements

This work was financially supported by the National Natural Science Foundation of China (52125201, 21975141, Y.Y.Z.) and the National Key Basic Research and Development Program (2020YFA0210702, Y.Y.Z.).

## Author contributions

Yingying Zhang (Y.Y.Z.) and H.L. conceived the idea and designed the experiments. Y.Y.Z. supervised the project. H.L. performed most of the experiments and analyzed all the data. Yong Zhang (Y.Z.) participated in wet spinning experiments. M.Z. contributed to code design. M.Z., S.L., and L.G. participated in the application demonstration of touching the perceptual textile to control the robot hand. H.L. and P.B. assisted in electrical tests. S.W., L.G., and H.W. participated in the preparation of electroluminescent devices. X.E.W. participated in the mechanical analysis. All authors discussed the results. H.L. and Y.Y.Z. co-wrote the manuscript with feedback from all authors.

## Competing interests

The authors declare no competing interests.
