## [Peer Review File · Nature Communications]

Reviewers' comments:

Reviewer #1 (Remarks to the Author):

This manuscript reports an innovative perceptual textile that is capable of detecting potential hazards and human touch, utilizing silk fibroin-based ionic hydrogel (SIH) fibers. The exceptional mechanical properties of SIH fibers, including a tensile strength of 55 MPa and an extensibility of 550%, coupled with a stable ionic conductivity of 0.45 S/m, make them superior to previously reported ionic conductive fibers. The SIH fiber-based textiles demonstrate good performance in detecting dangers and finger touch, and are expected to pave the way for novel skin-like textiles and human-machine interfaces. Overall, this work is fascinating and presents a significant advancement in the field of electronic textiles. I would like to recommend its publication after addressing the following specific comments:

1. As diameter is an important factor for practical application, the authors are suggested to discuss the factors that influence the diameter of SIH fibers during the preparation process and investigate whether the diameter can be tuned.
2. While the tensile properties of SIH fibers have been investigated through stress-strain curves, it is recommended to further record the cyclic stress-strain curves of SIH fiber to explore its cyclic tensile properties.
3. In line 184, it is stated that "when a drop of water falls on the SIH fiber, the current increases instantaneously and then gradually." The authors should clarify the reason for the three sudden increases in the curve shown in Figure 3d.
4. The authors designed a perceptual textile that can detect finger touch based on a SIH fabric and a surface-capacitive touch circuit system. It would be valuable to discuss the differences and advantages of the perceptual fabrics in this work compared to previously reported piezoresistive tactile textiles (such as *Nature Electronics*, 2021, 4(3): 193-201).
5. The caption of Figure 5f needs to be revised, as Figure 5f shows a schematic diagram of controlling various devices, not just robot hands, by touching the SIH textiles.

Reviewer #2 (Remarks to the Author):

After consideration of the study presented, it is acknowledged that the authors have developed a conductive silk fibroin-based ionic hydrogel (SIH) fiber with good mechanical properties and conductivity. The SIH fiber's semi-crystalline and oriented structure, along with the incorporation of [Emim]BF₄, enable it to exhibit excellent strength, extensibility, and stable conductivity. These properties make it a

promising material for the fabrication of functional units in the development of human-friendly intelligent wearable systems.

However, it is also noted that the concept of developing intelligent textiles capable of sensing potential dangers and detecting human touch based on ionic-conductive silk protein materials has been explored in previous literature, and the presented applications in this study might be too simple and limited in scope. While the ability to electrically respond to potential hazards and detect finger touch is useful, it might not present groundbreaking advancements in the field of wearable technology.

Therefore, based on the current information provided, the study might not be considered for publication in Nature Communications which require stronger innovation or more significant breakthrough. For comparison, some published works on ionic conductive silk fibroin materials for intelligent perception or wearable sensors are listed : " Silk Fibroin-Based Wearable All-Fiber Multifunctional Sensor for Smart Clothing" (Advanced Fiber Materials); "Self-Healing, Self-Adhesive Silk Fibroin Conductive Hydrogel as a Flexible Strain Sensor"(ACS APPLIED MATERIALS & INTERFACES); "Humanoid Ionotronic Skin for Smart Object Recognition and Sorting" (ACS Materials Letters); "Physically Cross-Linked Silk Fibroin-Based Tough Hydrogel Electrolyte with Exceptional Water Retention and Freezing Tolerance" (ACS Appl. Mater. Interfaces); "Transparent, highly-stretchable, adhesive, and ionic conductive composite hydrogel for biomimetic skin" (Journal of Materials Science)...

Reviewer #3 (Remarks to the Author):

The authors report an ionic hydrogel fiber made from silk fibroin with absorbing a mixture solvent of ionic liquid, glycerol and water. Applications of detecting finger touching, fire, and water are demonstrated to reveal the intelligence of ionic fibers. Even though the fibers showed remarkable mechanical properties, the material design strategies, manufacturing methods and applications are routine. Overall, this work does not show any new insight in the field. Hence, this reviewer believes that it cannot meet the criteria of Nat Commun. Other comments include:

1) More details on the solvent replacement process should be given. For example, what is the time duration of immersing fibers in the ionic liquid solution? What is the weight change before and after absorbing ionic solution, that is, how much solvent does the fibers absorb, and what is the dimension changes? Moreover, the detail composition of the ultimate ionic hydrogel fibers needs to be analyzed?

2) The 3.0M-S sample showed better thermal stability, but why describe "the 3.0M-S exhibited a higher weight loss than the controlled fibers"?

3) Why was the mechanical property of the ionic hydrogel fiber greatly enhanced when just incorporating a small amount of ionic liquid (0.1 M), while it was slightly increased with further increasing the content of ionic liquid.

4) What are changes in the dimension and morphology of the fibers after stretching treatment during the fabrication?

5) What types of samples are used for the conductivity test (Figure 3c), the ones prepared with stretching or not? In addition, the conductivity measurement details such as fiber size, electrode configuration, and electrical conditions should be given.

6) Also, why was the conductivity significantly enhanced when just incorporating a low amount of ionic liquid (0.1 M), while it just slightly increased with further increase of ionic liquid.

7) The explanation of swelling of hydrogel for the increase of current when exposing to water sounds non-reasonable. If so, what is the reason for the slight decrease of conductivity when storing for a long term? Is it because of absorbing water/moisture or dehydration? So, the improvement of ion transfer capacity needs to be detailed to support the claim.

8) The equivalent circuit for illustrating the finger touching mechanism is not reasonable. The resistance of the prepared ionic hydrogel fibers is much lower than that of the human body, so, how can an equivalent grounding be established through the finger?

9) How do you determine that the rise of conductivity when exposing to fire is caused by the carbonization of fibers, not just caused by the rise of temperature?

10) A comparison of the reported performance in terms of mechanical property and conductivity to reported similar materials is suggested.

Response to Reviewers' Comments

We thank the Reviewers for the careful review and thoughtful comments/suggestions for our manuscript. According to the comments, we have carefully revised the manuscript. The changes are marked in BLUE in the manuscript. The following is a point-to-point reply.

Reviewer #1:

Summary Comment:

This manuscript reports an innovative perceptual textile that is capable of detecting potential hazards and human touch, utilizing silk fibroin-based ionic hydrogel (SIH) fibers. The exceptional mechanical properties of SIH fibers, including a tensile strength of 55 MPa and an extensibility of 550%, coupled with a stable ionic conductivity of 0.45 S/m, make them superior to previously reported ionic conductive fibers. The SIH fiber-based textiles demonstrate good performance in detecting dangers and finger touch, and are expected to pave the way for novel skin-like textiles and human-machine interfaces. Overall, this work is fascinating and presents a significant advancement in the field of electronic textiles. I would like to recommend its publication after addressing the following specific comments:

Reply:

Thank you very much for your careful review and high evaluation on this work. The comments and suggestions have helped us to further improve our manuscript. According to your suggestions, we revised the manuscript carefully.

Comment #1:

1. As diameter is an important factor for practical application, the authors are suggested to discuss the factors that influence the diameter of SIH fibers during preparation process and investigate whether the diameter can be tuned.

Reply:

Thank you for your valuable suggestion on our work. The diameters of silk fibers are primarily influenced by the inside diameter of spinneret and the feeding rate of spinning dope. Consequently, we investigated the impact of these factors on the diameters of silk fibers. The results were shown in Figure S4. The diameter of silk fibers gradually reduced when decreasing the spinneret diameter and feeding rates.

According to your suggestion, we revised the manuscript and supplementary information.

For your convenience, we copied the related parts in the following:

In the manuscript:

(Page 6, Line 90) Besides, the diameter of silk fibers gradually reduced when decreasing the inside diameter of spinneret and the feeding rate (Figure S4).

(Page 19, Line 334) In order to investigate the factors influencing the diameters of fibers, we adjusted the spinneret diameters and feeding rates (Figure S4).

In the supplementary information:

(Page 3, Line 23)

Figure S4 Optical images of SIFs prepared using spinnerets with different diameters and feeding rates (The concentration of ionic liquid is 3M). (a) 20 G, $240 \mu\text{L}\cdot\text{min}^{-1}$; (b) 22 G, $180 \mu\text{L}\cdot\text{min}^{-1}$; (c) 24 G, $140 \mu\text{L}\cdot\text{min}^{-1}$.

Comment #2:

2. While the tensile properties of SIH fibers have been investigated through stress-strain curves, it is recommended to further record the cyclic stress-strain curves of SIH fiber to explore its cyclic tensile properties.

Reply:

Thank you for your important suggestion regarding this work. Accordingly, we recorded the cyclic stress-strain curves of silk fibroin-based ionic hydrogel (SIH) fibers (Figure S9). The SIH fibers showed obvious hysteresis after unloading of tension. For each sample, the loading-unloading cycles showed that little plastic deformation occurred in the SIH fibers after the first cycle, indicating the good fatigue resistance after the first cycle.

According to your suggestion, we revised the manuscript and supplementary information. For your convenience, we copied the related parts in the following:

In the manuscript:

(Page 10, Line 166) Besides, the SIH fibers showed obvious hysteresis during cyclic loading and unloading of tensile stress (Figure S9). The stress-strain curves show that little plastic deformation occurred in the SIH fibers after the first training cycle, indicating the good fatigue resistance.

In the supplementary information:

(Page 5, Line 49)

Figure S9 Cyclic tensile stress-strain curves with a tensile strain of 50% for twenty cycles of different SIH fiber samples. The samples are 0.0M-S (a), 0.1M-S (b), 0.5M-S (c), 1.0M-S (d), 2.0M-S (e), and 3.0M-S (f), respectively.

Comment #3:

3. In line 184, it is stated that “when a drop of water falls on the SIH fiber, the current increases instantaneously and then gradually.” The authors should clarify the reason for the three sudden increases in the curve shown in Figure 4d.

Reply:

Thank you for your careful review on this work. When a water droplet contacts with the SIH fiber, the output current of the fiber exhibits an abrupt surge, followed by a slow increase. The three distinct increases observed in Figure 4d can be attributed to the sequential deposition of three water droplets onto the fibers at different time points.

Accordingly, we have incorporated this information in the manuscript. For your convenience, we copied the related parts in the following:

In the manuscript:

(Page 13, Line 226) As shown in Figure 4d, three obvious steps were observed when three drops of water dripped in sequence.

Comment #4:

4. The authors designed a perceptual textile that can detect finger touch based on a SIH fabric and a surface-capacitive touch circuit system. It would be valuable to discuss the differences and advantages of the perceptual fabrics in this work compared to previously reported piezoresistive tactile textiles (such as Nature Electronics, 2021, 4(3): 193-201).

Reply:

Thank you for your suggestion on our work. The tactile interactive fabric (denoted as Textile-1) by Luo et al. stands as a remarkable accomplishment in the realm of human-environment interaction (Nature Electronics, 2021, 4(3): 193-201). In comparison, our intelligent fabric (termed as Textile-2) shows distinctive differentiations and merits:

(1) The fibers in Textile-1 consist of conductive iron cores ensheathed with copper/graphite coatings, whereas the fibers comprising Textile-2 are stretchable ion-conductive fibers. This inherent flexibility and stretchability imbue our fabric with a unique pliancy, ensuring both wearer comfort and flexibility.

(2) The sensing mechanisms employed by the two textiles diverge significantly. The sensory attribute of Textile-1 relies upon a resistive pressure approach, wherein a pair of orthogonal fibers coalesce to form a sensing unit. In contrast, our approach entails the integration of a surface capacitive sensing system through circuit design.

(3) The sensing modalities exhibited by the two textiles differ markedly. Textile-1 responds to any external pressure stimulus, lacking the specificity required to discern human touch interactions. In contrast, our fabric selectively responds solely to human touch, effectively disregarding mechanical deformations induced by external objects or pressure.

(4) Operational disparity between the textiles is also evidenced in their wiring necessities. While the efficacy of Textile-1 hinges upon an intricate network of external conductive pathways, our textile minimizes such requirements, necessitating only minimal external wiring for seamless functionality.

In essence, we used stretchable ionic conductive fibers to prepare the present flexible textile for use as a novel human-machine interface.

Accordingly, we incorporated related information into the manuscript. For your convenience, we copied the related parts in the following:

In the manuscript:

(Page 16, Line 287) Compared to previous tactile textiles¹⁵, our perceptual textile shows distinctive differentiations and merits. Firstly, the fibers comprising our textiles are stretchable ionic conductive fibers rather than stiff iron wires, endowing our textiles with inherent flexibility. Secondly, we realize the perceptual function by developing a surface capacitive sensing system instead of piezoresistive principle. Thirdly, our textiles respond selectively only to human touch, not to any external pressure stimulus. Lastly, our perceptual textiles minimize wiring requirements and eliminate the need for complex external circuits. In summary, our textile obviously differs from previous tactile textiles, demonstrating great application potential as a textile-type human-machine interface.

Comment #5:

5. The caption of Figure 5f needs to be revised, as Figure 5f shows a schematic diagram of controlling various devices, not just robot hands, by touching the SIH textiles.

Reply:

Thank you for your careful review on our work. Accordingly, we have revised the caption of Figure 5f to “(f) Diagram of controlling various devices by touching a SIH fiber-based perceptual textile.” For your convenience, we copied the related parts in the following:

In the manuscript:

(Page 15, Line 248) Figure 5 (f) Diagram of controlling various devices by touching a SIH fiber-based perceptual textile.

Reviewer #2:**Comment Part I:**

After consideration of the study presented, it is acknowledged that the authors have developed a conductive silk fibroin-based ionic hydrogel (SIH) fiber with good mechanical properties and conductivity. The SIH fiber's semi-crystalline and oriented structure, along with the incorporation of [Emim]BF₄, enable it to exhibit excellent strength, extensibility, and stable conductivity. These properties make it a promising material for the fabrication of functional units in the development of human-friendly intelligent wearable systems.

Reply:

Thank you for your high evaluation on our fibers, encompassing their exceptional mechanical properties, stable conductivity, and potential as human-friendly components in intelligent wearable systems. We agree with you that it is a promising material for the fabrication of functional units in the development of human-friendly intelligent wearable systems.

Comment Part II:

However, it is also noted that the concept of developing intelligent textiles capable of sensing potential dangers and detecting human touch based on ionic-conductive silk protein materials has been explored in previous literature, and the presented applications in this study might be too simple and limited in scope. While the ability to electrically respond to potential hazards and detect finger touch is useful, it might not present groundbreaking advancements in the field of wearable technology.

Therefore, based on the current information provided, the study might not be considered for publication in Nature Communications which require stronger innovation or more significant breakthrough. For comparison, some published works on ionic conductive silk fibroin materials for intelligent perception or wearable sensors are listed : " Silk Fibroin-Based Wearable All-Fiber Multifunctional Sensor for Smart Clothing" (Advanced Fiber Materials); "Self-Healing, Self-Adhesive Silk Fibroin Conductive Hydrogel as a Flexible Strain Sensor"(ACS APPLIED MATERIALS & INTERFACES); "Humanoid Ionotronic Skin for Smart Object Recognition and Sorting" (ACS Materials Letters); "Physically Cross-Linked Silk Fibroin-Based Tough Hydrogel Electrolyte with Exceptional Water Retention and Freezing Tolerance" (ACS Appl. Mater. Interfaces); "Transparent, highly-stretchable, adhesive, and ionic conductive composite hydrogel for biomimetic skin" (Journal of Materials Science) ...

Reply:

Thank you for your careful review and suggestions. We are sorry for failed to convince you the novelty and significance of this work in the original version. Taking your input into account, we further highlighted the advances and novelty of our work in the revised version. In short, we designed and fabricated silk hydrogel fibers with intrinsic ion conductivity, excellent mechanical properties, which catalyzed the functionalization and intelligent utilization of silk fibers. More importantly, we achieved perceptual textiles capable of exclusively responding to human touch and accurately pinpointing their location, which is expected to revolutionize human-machine interfaces, promising great convenience in the realm of human intelligent living.

Thank you for providing information about some related work. Accordingly, we carefully read and analyzed the literature you mentioned. Compared to these published studies, we believe our work stands out distinctly and shows unexplored innovations. The following is a detailed analysis:

(1) Journal: *Advanced Fiber Materials* (2022, 4, 873)

Title: **Silk Fibroin-Based Wearable All-Fiber Multifunctional Sensor for Smart Clothing**

It reported a silk membrane prepared through electrospinning then applying Ag interdigital electrodes on the surface for capacitance sensing. **It is a nonwoven silk nanofiber film** rather than a textile, and **it also lacks inherent conductivity**. Furthermore, the sensor cannot match the precision of our SIH fiber-based plain weave fabric in accurately pinpointing the locations of human touch.

(2) Journal: *ACS Applied Materials & Interfaces* (2021, 13, 40013)

Title: **Self-Healing, Self-Adhesive Silk Fibroin Conductive Hydrogel as a Flexible Strain Sensor**

The authors designed ion-conductive hydrogel bulks by proportionally mixing polyvinyl alcohol, silk fibroin, and borax. Clearly, **the material is a hydrogel rather fibers or textiles**, let alone being utilized as a comfortable electronic textile.

(3) Journal: *ACS Materials Letters* (2022, 5, 189)

Title: **Humanoid Ionotronic Skin for Smart Object Recognition and Sorting**

This work reported **an ionic membrane made from bacterial nanocellulose, silk fibroin, and calcium chloride**, followed by assembly with acrylic acid elastomer to produce a triboelectric generator. It's evident that this does not correspond to ion-conductive silk fibers or fabrics, and it does not enable precise recognition of human touch and touch locations for human-machine interaction.

(4) Journal: *ACS Applied Materials & Interfaces* (2020, 12, 25353)

Title: **Physically Cross-Linked Silk Fibroin-Based Tough Hydrogel Electrolyte with Exceptional Water Retention and Freezing Tolerance**

This work reports a hydrogel made through the incubation of ion liquids, silk fibroin, and water, which were then injected into silicone tubes to create a segment of string. Clearly, this differs a lot from our continuous and rapid fabrication of ion-conductive silk fibers for textile-type human-machine interfaces.

(5) Journal: *Journal of Materials Science* (2021, 56, 2725)

Title: **Transparent, highly-stretchable, adhesive, and ionic conductive composite hydrogel for biomimetic skin**

The authors combined polyacrylamide, silk sericin, and sodium chloride to fabricate ion gel bulks or films, recording the resistance changes under various strains. It's evident that this differs significantly in composition, constitution, and performance from our fibroin-based ion-conductive fibers. Moreover, it doesn't explore the innovative applications in aiding robots to perceive external dangers or assisting humans as textile-based human-machine interfaces.

Based on the above information, you will see that our work differs a lot with the published ones. Recent advances in electronic textiles have enabled communication, sensing, display, power supply, and other functions (*Nature* 2018, 560, 214; *Nature* 2019, 569, 698; *Nature* 2021, 597, 57; *Nature* 2021, 591, 240). And our research has achieved a pivotal function in textile-based human-machine interaction, poised to play a crucial role in human intelligent living.

According to your comments and suggestion, we further highlighted the novelty and significance of this work in the revised manuscript. For your convenience, we copied the related parts in the following:

In the manuscript:

(Page 3, Line 48) In this work, we prepared a highly strong, conductive, and stable silk fibroin-based ionic hydrogel (SIH) fiber and realized the fabrication of an intelligent perceptual textile capable of precisely detecting external dangers and human touch.

(Page 4, Line 61) Therefore, the successful fabrication of silk fibers with intrinsic ion conductivity and excellent mechanical properties will promote a significant advancement in the functionalization and utilization of silkworm silk fibers. Furthermore, the development of perceptual textiles capable of exclusively and accurately pinpointing human touch will revolutionize human-machine interfaces, offering excellent flexibility and comfort, bringing great convenience in intelligent living to human.

Reviewer #3:**Summary Comment:**

The authors report an ionic hydrogel fiber made from silk fibroin with absorbing a mixture solvent of ionic liquid, glycerol and water. Applications of detecting finger touching, fire, and water are demonstrated to reveal the intelligence of ionic fibers. Even though the fibers showed remarkable mechanical properties, the material design strategies, manufacturing methods and applications are routine. Overall, this work does not show any new insight in the field. Hence, this reviewer believes that it cannot meet the criteria of Nat Commun.

Other comments include:

Reply:

Thank you very much for your careful review on our work. According to your suggestion, we highlighted the novelty and significance of this work in the revised manuscript. In our study, we designed and fabricated silk hydrogel fibers with intrinsic ion conductivity, exceptional tensile strength, and excellent stability. More importantly, we designed and developed textile-type human-machine interfaces capable of exclusively responding to human touch and accurately pinpointing their location, contributing to unparalleled convenience in the context of intelligent living. In the current landscape, intelligent textiles have already demonstrated functionalities encompassing communication, sensing, display, and power supply (Nature 2018, 560, 214; Nature 2019, 569, 698-702; Nature 2021, 597, 57; Nature 2021, 591, 240). Our innovative textile that can be used as human-machine interface is poised to propel the ongoing evolution of intelligent textiles. Taking your recommendations into account, we revised the manuscript carefully. For your convenience, the relevant content was copied below:

In the manuscript:

(Page 3, Line 48) In this work, we prepared a highly strong, conductive, and stable silk fibroin-based ionic hydrogel (SIH) fiber and realized the fabrication of an intelligent perceptual textile capable of precisely detecting external dangers and human touch.

(Page 4, Line 61) Therefore, the successful fabrication of silk fibers with intrinsic ion conductivity and excellent mechanical properties will promote a significant advancement in the functionalization and utilization of silkworm silk fibers. Furthermore, the development of perceptual textiles capable of exclusively and accurately pinpointing human touch will revolutionize human-machine interfaces, offering excellent flexibility and comfort, bringing great convenience in intelligent living to human.

Your detailed comments and suggestions highly improved our manuscript. The following is point-to-point reply to each of your comments.

Comment #1:

1) More details on the solvent replacement process should be given. For example, what is the time duration of immersing fibers in the ionic liquid solution? What is the weight change before and after absorbing ionic solution, that is, how much solvent does the fibers absorb, and what is the dimension changes? Moreover, the detail composition of the ultimate ionic hydrogel fibers needs to be analyzed?

Reply:

Thank you for your meticulous review on our research. Accordingly, we carried out a series of supplementary experiments, characterizations, and analyses. These endeavors led to an improvement of our work. The detailed replies are as follows:

(a) Duration time

The time duration of immersing fibers in the ionic liquid solution is 30 min.

(b) Weight change

The weight per unit length of various fibers was measured and the data was shown in Figure S5a. The red dashed line represents the weight ($0.6 \text{ g}\cdot\text{m}^{-1}$) of bare regenerated silk. Compared to bare regenerated silk, the weight variations of silk fibroin-based ionic hydrogel (SIH) fibers (From 0.0M-S to 3.0M-S) were 188%, 224%, 241%, 254%, 277%, and 295%, respectively (Figure S5c). This result elucidates a progressive trend in unit-length weight of SIH fibers as the ionic liquid concentration increases.

(c) Dimension change

The morphological and diameter of bare regenerated silk fibers were observed through optical microscopy. Figures S3 shows their surface and cross-sectional morphology, respectively. The diameter of bare regenerated silk fibers was $269 \mu\text{m}$. As shown in Figure S2, in comparison to bare regenerated silk fibers, the diameter of SIH fibers (From 0.0M-S to 3.0M-S) increased by 49.8%, 54.9%, 58.4%, 61.4%, 69.4%, and 71.4%, respectively.

(d) Composition analysis of SIH fibers

Figure S6 and Table S1 show the ratios of ionic liquid, water, and glycerol within ionic liquid solutions. According to the content of fibroin and solvent within the different SIH fibers (Figure S2), the detailed composition of SIH fibers was calculated and the data was

shown in Figure S7 and Table S2. The composition of the 0.0M-S exclusively comprises silk fibroin, glycerol, and water, accounting for 34.7%, 36.9%, and 28.4% respectively. An escalation in the concentration of ionic liquid in the immersion solution yields an increment in the ionic liquid content within SIH fibers. This content scales to 1.2%, 6.1%, 12.3%, 24.8%, and 37.2% in the ascending order of ionic liquid concentrations. Concomitantly, the ratios of silk fibroin, glycerol, and water within these fibers exhibit a diminishing trend as the ionic liquid content amplifies. Silk fibroin's proportion decreases progressively to 30.8%, 29.3%, 28.2%, 26.5%, and 25.3%. Glycerol accounts for 38.4%, 36.5%, 33.6%, 27.5%, and 21.2%. Water occupy 29.6%, 28.1%, 25.9%, 21.2%, and 16.3%. These experimental results and analyses were incorporated into the manuscript and supplementary information. For your convenience, we copied the related parts in the following:

In the manuscript:

(Page 6, Line 90) Besides, the diameters of all SIF fibers are greater than that of bare regenerated silk fibers not subjected to solvent exchange within the ionic liquid reservoir (Figure S2-S3), indicating that the treatment of fibers within ionic liquid solution improves their diameters. In addition, we can tune the fiber diameter by adjusting the spinneret diameters and feeding rates (Figure S4). We also quantified the weight variation during the solvent exchange process within the ionic liquid reservoir (Figure S5). The data elucidates a progressive trend in unit-length weight as the ionic liquid concentration increases. Subsequently, according to the composition of ionic liquid solution (Figure S6 and Table S1) and the content of fibroin and solution within different SIH fibers (Figure S2), the detailed composition of SIH fibers

was calculated and the data was shown in Figure S7 and Table S2. An escalation in the concentration of ionic liquid yields an increment in the ionic liquid content within SIH fibers. Concomitantly, the ratios of silk fibroin, glycerol, and water within these fibers exhibit a diminishing trend as the ionic liquid content amplifies.

In the supplementary information:

(Page 2, Line 15)

Figure S2 Diameters of SIH fibers. (a) Diameter of fibers in different groups. The red dashed line signifies the diameter of bare regenerated silk fibers ($D_0=269 \mu\text{m}$). (b) Diameter variations of the different fiber groups in comparison to bare regenerated silk fibers.

Figure S3 Morphology and dimensions of bare regenerated silk fibers. (a) Surface. (b) Cross-section.

(Page 3, Line 29)

Figure S5 The weight change of silk fibers during the solvent process in ionic liquid tank. (a) Weight of fibers per unit length. The red dashed line represents the weight per unit length ($W_0=0.6 \text{ g}\cdot\text{m}^{-1}$) of bare regenerated silk. (b) Solvent contents in unit-length fibers. (c) Weight variations of SIH fibers prepared with varied ionic liquid concentrations in comparison to bare regenerated silk.

Figure S6 Component content in ionic liquid solutions with different concentrations

Figure S7 Contents of different components in various SIH fibers.

(Page 8, Line 78)

Table S1 Composition of ionic liquid solutions with varied concentrations

Solution	Weight fraction (%)		
	Ionic liquid	Glycerol	Water
0.0 M	0.0	56.5	43.5
0.1 M	1.7	55.5	42.8
0.5 M	8.6	51.6	39.8
1.0 M	17.2	46.8	36.1
2.0 M	33.7	37.4	28.9
3.0 M	49.8	28.3	21.8

Table S2 Composition of SIH fibers fabricated with different concentrations of ionic liquid solutions.

Sample	Weight fraction (%)			
	Silk fibroin	Ionic liquid	Glycerol	Water

0.0 M-S	34.7	0.0	36.9	28.4
0.1 M-S	30.8	1.2	38.4	29.6
0.5 M-S	29.3	6.1	36.5	28.1
1.0 M-S	28.2	12.3	33.6	25.9
2.0 M-S	26.5	24.8	27.5	21.2
3.0 M-S	25.3	37.2	21.2	16.3

Comment #2:

2) The 3.0M-S sample showed better thermal stability, but why describe “the 3.0M-S exhibited a higher weight loss than the controlled fibers”?

Reply:

Thank you for pointing out this issue. We are sorry for the mistake and revised the main text. For your convenience, we copied the related parts in the following:

In the manuscript:

(Page 7, Line 120) Furthermore, the 3.0M-S exhibited a lower weight loss than the controlled fibers (0.0M-S) in the temperature range of 30°C–375°C, indicating improved thermal stability of SIH fibers upon incorporating [Emim]BF₄.

Comment #3:

3) Why was the mechanical property of the ionic hydrogel fiber greatly enhanced when just incorporating a small amount of ionic liquid (0.1 M), while it was slightly increased with further increasing the content of ionic liquid.

Reply:

Thank you for your insightful question. As depicted in the Figure S7, with the introduction of a small amount of ionic liquid (0.1 M), the ratio of glycerol and water to silk fibroin within the fiber notably escalates. However, as the concentration is further increased, this ratio gradually diminishes. Glycerol and water act as plasticizers within the fiber, mitigating crack propagation along the radial axis during stretching, thus reducing brittleness in fiber fracture. Consequently, the mechanical performance of 0.1M-S fibers attains its optimum. Subsequent elevation of the ionic liquid concentration, resulting in a decreased glycerol and water to silk fibroin ratio, leads to a progressive deterioration in fiber mechanical properties. Furthermore, significant diameter augmentation of fibers becomes evident with substantial concentration increments (as illustrated in Figure S2), coinciding with a marked reduction in crystal orientation within the fiber (as depicted in Figure 2f). These combined factors contribute to the diminishing mechanical performance of fibers with increasing ionic liquid concentration. In response to your comments, we added these analyses into the manuscript. For your convenience, we copied the related parts in the following:

In the manuscript:

(Page 10, Line 157) We further studied the influence of ionic liquid concentration on the mechanical properties of the SIH fibers. With the introduction of a small amount of ionic liquid (0.1 M), the content of glycerol and water to silk fibroin within the fiber notably escalates (Figure S7). However, as the concentration of ionic liquid increases further, both the content of glycerol and water decrease. Glycerol and water act as plasticizers within the fiber, mitigate crack propagation along the radial axis

during stretching, thus reducing brittleness of fibers. Subsequent elevation of the ionic liquid concentration decreased the content ratios of glycerol, water, and silk fibroin, and increased the diameter of silk fibers, leading to a progressive deterioration in the mechanical properties of fibers.

Comment #4:

4) What are changes in the dimension and morphology of the fibers after stretching treatment during the fabrication?

Reply:

Thank you for your careful review on our work. Accordingly, we carried out further experiments and analyses. As shown in Figure S10, the diameter of SIH fiber gradually decrease as the stretching ratio increases from 0% to 200%. The diameters of the fibers are 415 μm , 376 μm , 319 μm , 295 μm , and 235 μm , respectively. Optical microscopy examinations reveal that the silk fibers possess uniform diameters and smooth surfaces.

We added these results and discussions into the manuscript and supplementary information. For your convenience, we copied the related parts in the following:

In the manuscript:

(Page 10, Line 172) The diameters of fibers gradually decrease as the stretching ratio increases from 0% to 200% (Figure S10). The silk fibers in all groups possess uniform diameters and smooth surfaces (Figure S11).

In the supplementary information:

(Page 6, Line 54)

Figure S10 Diameter variations of SIH fibers with varied stretching ratios.

Figure S11 Optical images of SIH fibers prepared with varied stretching ratios. (a) 0%; (b) 50%; (c) 100%; (d) 150%; (e) 200%.

Comment #5:

5) What types of samples are used for the conductivity test (Figure 3c), the ones prepared with stretching or not? In addition, the conductivity measurement details such as fiber size, electrode configuration, and electrical conditions should be given.

Reply:

Thank you for pointing out the missing of these experimental details. The samples without being stretched were utilized for testing conductivity. During the electrical performance evaluation, the gauge length of SIH fibers is 20 mm, with their ends connected to conductive wires through silver conductive paste. The *i-t* curves of SIH fibers were

measured using CHI760E electrochemical workstation. The testing procedure entailed an initial voltage of 1 V, a sample interval of 0.1 s, and a run time of 30 s.

Accordingly, we added related information into the manuscript. For your convenience, the related parts are copied in the following:

In the manuscript:

(Page 20, Line 366) The samples utilized for testing conductivity were the SIH fibers without being stretched. During the electrical performance evaluation, the gauge length of SIH fibers is 20 mm, with their ends connected to conductive wires through conductive silver paste. The *i-t* curves of various SIH fibers were measured. The testing procedure entailed an initial voltage of 1 V, a sample interval of 0.1 s, and a run time of 30 s.

Comment #6:

6) Also, why was the conductivity significantly enhanced when just incorporating a low amount of ionic liquid (0.1 M), while it just slightly increased with further increase of ionic liquid.

Reply:

Thank you for your careful review regarding this work. As illustrated in Figure 3c, the conductivity of SIH silk fibers demonstrates an almost linear relationship as the concentration of ionic liquid rises from 0.0 M to 1.0 M, which is attributed to the rapid increase in the ionic liquid content of the SIH fibers (Figure S7). As the concentration surpasses 2.0 M, the rate of increase in conductivity for the ionic gel fibers decreases, which

is ascribed to that the enhanced interaction between oppositely charged ions in high concentration of ionic liquid solutions hampers ion mobility.

As depicted in Figure S7, the concentration of the ionic liquid within the fibers progressively increases. Furthermore, at ionic liquid concentrations ≤ 1.0 M, there is a notable surge in the concentration of the ionic liquid within the SIH fibers, with the ionic liquid content being lower than that of the solvent water. Conversely, at ionic liquid concentrations exceeding 1.0 M, despite the elevated concentration of the ionic liquid, its content surpasses that of the solvent water. Ionic liquids are considered strong electrolytes. For such compounds, in low concentration ranges, the low interionic attractions and high ion mobility lead to significant increases in conductivity with increasing concentration. However, as the concentration further increases, the enhanced interaction between oppositely charged ions hampers ion mobility, slowing down the rate of increase in conductivity.

Accordingly, we revised the manuscript. For your convenience, we copied the related parts in the following:

In the manuscript:

(Page 11, Line 180) As shown in Figure 3c, from 0.0M-S to 1.0 M-S, the conductivity progressively increases, which is attributed to the rapid increase in the ionic liquid content of the SIH fibers (Figure S7). As the concentration reaches 2.0 M, the rate of increase in conductivity of the SIH fibers decreases, which is ascribed to that the enhanced interaction between oppositely charged ions in high concentration of ionic liquid solutions hampers ion mobility.

Comment #7:

7) The explanation of swelling of hydrogel for the increase of current when exposing to water sounds non-reasonable. If so, what is the reason for the slight decrease of conductivity when storing for a long term? Is it because of absorbing water/moisture or dehydration? So, the improvement of ion transfers capacity needs to be detailed to support the claim.

Reply:

Thank you for bringing out these questions. In response to your inquiries, we conducted an analysis of the pertinent questions.

The conductivity of the fiber can be determined by the following equation:

$$\sigma = 4 \times 10^{10} \times L / (\pi D^2 R)$$

where L (cm) and D (μm) are the length and diameter of the fiber, respectively. R (Ω) represents the fiber resistance which is obtained through the following equation:

$$R = U / I$$

where U (V) is the applied voltage across the fiber (1 V), and I (A) is the current passing through the fiber.

Through the utilization of the aforementioned equations, we deduce:

$$I = \pi D^2 U \sigma / (4 \times 10^{10} \times L)$$

Letting $k = \pi U / (4 \times 10^{10} \times L)$, where U and L are constants, implies k is a constant as well.

Consequently:

$$I=kD^2\sigma$$

Considering that the addition of water droplets diminishes the ionic liquid concentration within the fiber, thus reducing σ . I concurrently increased, which indicates an augmentation in D . This result illustrates that the fiber undergoes swelling subsequent to the water droplet addition.

These analyses indicate that, although the conductivity of the fiber decreases upon the addition of water droplets, the cross-sectional area of the fiber increases. The amplification of I by the increased area far outweighs the diminishing effect of conductivity decrease on I , leading to a net increase in the total I of the fiber.

Regarding the minor reduction in conductivity during long-term storage, we attribute this to a gradual dehydration effect induced by the prolonged exposure in an environment with a low RH around 43%.

Accordingly, we added related discussion into the manuscript. For your convenience, we copied the related parts in the following:

In the manuscript:

(Page 11, Line 190) This minor reduction in conductivity during long-term storage is attributed to a gradual dehydration effect induced by the prolonged exposure.

(Page 13, Line 229) The current can be determined by the following equation:

$$I=\pi D^2 U \sigma / (4 \times 10^{10} \times L)$$

Where D (μm) and L (cm) are the diameter and length of the fiber, respectively. U (V) is the applied voltage on the fiber (1 V), I (A) and σ are the output current and conductivity of the fiber. The addition of water droplets diminishes the ionic liquid

concentration, thus reducing σ . I concurrently increased, which indicates an augmentation in D . That is, the amplification of current by the increased area far outweighs the diminishing effect of conductivity decrease on current, leading to a net increase in the output current of the fiber.

Comment #8:

8) The equivalent circuit for illustrating the finger touching mechanism is not reasonable. The resistance of the prepared ionic hydrogel fibers is much lower than that of the human body, so, how can an equivalent grounding be established through the finger?

Reply:

Thank you for your meticulous review on our work. Accordingly, we conducted further analyses on the equivalent circuits. We elucidate this question using the single SIH fiber as an example (Figure 5b). In practical application, the fiber length utilized is approximately 10 cm to facilitate the customization of the corresponding human-machine interface and to accommodate finger touch interactions.

When employing 2.0M-S for the fabrication of touch-sensitive textiles, the fiber resistance can be calculated to be approximately 1.363 M Ω . The calculation process is as follows:

$$R = L/(\sigma S) = 4L/(\pi\sigma D^2)$$

where $L = 0.1$ m, $D = 4.556 \times 10^{-4}$ m, and $\sigma = 0.45$ S·m⁻¹. This yields $R = 1.363$ M Ω .

Human body resistance can be measured using a multimeter, as depicted in Figure R1. When a person holds the two electrodes of the multimeter with their hands, the body's resistance is measured as ~ 1.491 M Ω .

Figure R1 The resistance of human body.

This resistance value might vary due to changes in environmental humidity or bodily conditions. However, it remains relatively close to the resistance of the ionic touch-sensitive textiles.

Furthermore, the determination of touch position does not involve human body resistance and can be obtained through the following formula:

$$x = \frac{R_0 + R - kR_0}{(k + 1)R}$$

Where R_0 and R denote the resistances of the fixed resistor (1 M Ω) and SIH fiber, respectively. The parameter k is defined as:

$$k = U_1/U_2$$

Here, U_1 and U_2 represent the voltages across the left and right fixed resistors in Figure 5b, respectively, which can be acquired from electrical testing instruments.

Throughout testing and practical utilization, the SIH fiber-based textiles can be worn on individuals. The ground electrode of the alternating current power is connected to the skin. When we touch the ionic gel fibers on the fabric using finger, a circuit forms. Ultimately, the accurate pinpointing of human touch is realized by the design and implementation of

the perceptual fabric. Accordingly, we added this information into the manuscript. For your convenience, we copied the related parts in the following:

In the manuscript:

(Page 15, Line 256) When the SIH fiber-based textiles are worn on the human body, the ground electrode of the power is connected to the skin. A closed circuit is formed when a finger touches the fiber.

Comment #9:

9) How do you determine that the rise of conductivity when exposing to fire is caused by the carbonization of fibers, not just caused by the rise of temperature?

Reply:

Thank you for your insightful question. We carried out further experiments to investigate the reasons for the current increase of fibers when exposing to fire. In Figure 4, the temperature of the fire was 981 ° C (Figure R2). As shown in Figure 2a, the SIH fibers were decomposed and carbonized at this temperature. Therefore, the rise of current in Figure 4c can be attributed to the rise of temperature and the carbonization of fibers.

Figure R2 The temperature of the fire in Figure 4a.

Based on these findings, we revised the manuscript. For your convenience, we have included the relevant excerpt below:

In the manuscript:

(Page 13, Line 221) The temperature of the fire is 981 °C. According to Figure 2a, the SIH fibers were decomposed and carbonized at this temperature. Therefore, the rise of current in Figure 4c is attributed to the rise of temperature and the carbonization of SIH fiber. And the decrease in current to zero is ascribed to the failure of SIH fiber.

Comment #10:

10) A comparison of the reported performance in terms of mechanical property and conductivity to reported similar materials is suggested.

Reply:

Thank you for your kindly suggestions. Accordingly, we compared the mechanical and electrical performance of our SIH fibers with reported ionic conductors. The results are supplied as Figure S13 and Table S3 in supporting information. As depicted in Figure S13a, in contrast to various other ionic conductive materials, our fibers exhibit higher strength and moderate elongation at break, effectively addressing the existing gaps in the mechanical properties within the related domain. Moreover, our fibers with stretching boasts an absolute mechanical strength advantage over a spectrum of ionic conductive materials (Figure S13b). A holistic assessment of the mechanical and electrical attributes, as showcased in Figure S13c, underscores the fact that our fibers combine heightened strength with a good conductivity. With exceptional mechanical and electrical properties, our fibers are poised to play an important role in the realm of smart textiles. Accordingly,

we revised the manuscript and supplementary information. For your convenience, we copied the related parts in the following:

In the manuscript:

(Page 11, Line 197) Performance comparison proves that our fibers have excellent mechanical and electrical properties (Figure S13 and Table S3). In contrast to various other ionic conductive materials, our fibers exhibit higher strength and moderate elongation at break, effectively addressing the existing gaps in the mechanical properties within the related domain (Figure S13a). Besides, our fiber with stretching boasts an absolute mechanical strength advantage over a spectrum of ionic conductive materials (Figure S13b). Figure S13c underscores that our fibers combine heightened strength with a moderate level of conductivity, effectively bridging the existing gaps in the relevant domain. With exceptional mechanical and electrical properties, our fibers are poised to play an important role in the realm of smart textiles.

In the supplementary information:

(Page 7, Line 67)

Figure S13 Performance comparison of SIH fibers with ionic conductors. (a) Mechanical comparison between SIH fibers and other ionic conductive materials. (b) Mechanical comparison between stretched SIH fibers and other ionic conductive

materials. (c) Comparison of combined mechanical and electrical performance between SIH fibers and other ionic conductors.

(Page 9, Line 84)

Table S3 The mechanical and electrical properties of ionic conductors in Figure S13

Name	Tensile strength (MPa)	Elongation at break	Conductivity (S·m⁻¹)	References
Ionic conductor (Liquid free)	0.09	1.5	0.2	1
	0.42	1.37	0.025	2
	0.24	11	1E-07	3
	0.15	11	0.0024	4
	0.12	0.6	0.00801	5
	0.32	11.5	0.000001	6
	0.3	4.5	0.01	7
	0.24	5.4	0.0131	8
	0.11	1.45	0.001	9
	1.5	16.4	0.001	10
Ionic conductor	1.3	5.2	2.6	11
	0.752	25.8	0.00169	12
	0.28	1.2	7.49	13
	0.2	4	0.765	14
	2.5	2.15	2.1	15
	3.525	0.8233	0.03	
	2.09	0.8308	0.017	16
	0.854	0.6466	-	

REVIEWER COMMENTS

Reviewer #1 (Remarks to the Author):

This work reported an ionic-conductive silk fiber. Although the work has undergone detailed analysis and characterization, I believe that all of its work is quite basic. In addition, there are many core issues that have not been explored in this work. In summary, the work has not yet reached the level of this journal at the current level.

1. To my knowledge, similar ionic silk fibers, similar functional properties with high conductivity and strength, and similar application displays for perception have been widely studied and reported. What are the novelty or different aspects of this research work worth publishing in this journal?
2. The biggest problem of ionic hydrogel is dehydration, which affects the overall stability of the device. Although the authors have made long-term storage tests on the conductivity of the SIH fiber, it cannot meet such high humidity conditions (43% humidity) in practical applications. Therefore, it is recommended that the authors supplement the stability of the SIH fiber under actual environmental conditions to evaluate its actual use efficiency.
3. Regarding the comparison with previous tactile textiles, the authors propose four merits. However, I have a different perspective. Firstly, commercial silver plated nylon conductive yarns also have excellent conductivity, excellent mechanical strength and ductility, and relatively low critical costs. Secondly, piezoelectric or triboelectric modes seem to have more advantages as active electrical response mechanisms compared to passive capacitive modes. Thirdly, I don't understand the difference between human contact and external pressure stimulation, they can be regarded as the same. Lastly, conductive connections are still required between SIH fibers of different channels, how to reduce external circuits. The author's final highlight is completely unconvincing.
4. It is necessary to characterize the distribution form and state of ionic liquids in the SIH fibers through experiments.
5. In Figure 1, it can be seen that ionic liquids are coated on the surface of silk fibroin fibers through infiltration, but there is not a strong interaction between the two. Therefore, how to maintain stability between the two interfaces. Ionic liquids are on the surface and their stability is more difficult to ensure. It is recommended to place ionic liquids on the core layer more appropriately.

Reviewer #2 (Remarks to the Author):

The concerns of the reviewers are well addressed, and the manuscript has been well revised. I recommend an acceptance of this article by Journal of Nature communications.

Reviewer #3 (Remarks to the Author):

I appreciate the hard work of the authors. They have addressed my comments and I have no further comment.

Response to Reviewer #1's Comments

Reviewer #1:

Summary Comment:

This work reported an ionic-conductive silk fiber. Although the work has undergone detailed analysis and characterization, I believe that all of its work is quite basic. In addition, there are many core issues that have not been explored in this work. In summary, the work has not yet reached the level of this journal at the current level.

Reply:

Thank you for the careful review and thoughtful comments on our revised manuscript. We also thank you for appreciating the detailed analysis and characterization of this work. As for the novelty and significance of this work, we will reply in the following specific points. On the whole, we have re-revised the manuscript again according to your new points. The changes are marked in BLUE in the manuscript. The following is a point-to-point reply. With this, we hope you will find the manuscript is further improved and can fulfill the requirement for publication in Nature Communications. However, if you have any further concerns or questions, we would be more than happy to address them. Thank you for your patience.

Comment #1:

To my knowledge, similar ionic silk fibers, similar functional properties with high conductivity and strength, and similar application displays for perception have been widely studied and reported. What are the novelty or different aspects of this research work worth publishing in this journal?

Reply:

Thank you for taking the time to review and share your opinion on the novelty of this work, which provided valuable direction for further improvement. The novelty of this work can be further explained as follows:

The advance and novelty of our research includes: (a) A **stretchable silk fibroin-based hydrogel fiber with excellent ionic conductivity and mechanical properties** was designed and fabricated. This is the first report on preparing ionic conductive silk fibers containing ionic liquids through wet spinning, which has been further fabricated into plain-weave ionic silk textiles. (b) Furtherly, we designed two kinds of perceptual textiles through designing two kinds of circuits. The first type of textile is based on an ordinary resistance response mode (Figure S20), and can recognize external stimuli such as fire burning, water immersing, and sharp object cutting (Figure 4). More importantly, the second kind of textile is based on a surface capacitive sensing system (Figure 5b and 5d). **This system makes our textile respond only to the touch of human hands, but has no electrical response to the press of other objects.** This attribute arises from the textile's ability to establish a closed circuit upon contact with a human finger (Figure 5b and 5d), and it does not respond to external pressure (Figure 3d). **This feature allows it to be used as a convenient and comfortable textile-type human-machine interface to help people interact with other machines by touching this textile, while avoiding the interference from contact with other objects.** This stands in contrast to existing tactile textiles, which respond to the pressure from various sources because of their response mechanism and are only suitable to detect pressure from other external objects.

Recently, various types of functional textiles have been developed, including communication, display, sensing, energy, etc. (*Nature* 2018, 560, 214; *Nature* 2019, 569,

698; *Nature* 2021, 597, 57; *Nature* 2021, 591, 240). **Our work provides a novel textile capable of precisely recognizing human touch and accurately pinpointing its location, and it is obviously different from other perceptive textiles based on pressure sensors.**

Accordingly, we further revised the manuscript to underscore the novelty and significance of this work. For your convenience, the related parts are copied in the following:

In the manuscript:

(Page 2, Line 22) Additionally, we designed perceptual textiles that can precisely and specifically recognize and pinpoint finger touch, serving as convenient and accurate human-machine interfaces.

(Page 3, Line 36) However, perceptual textiles capable of exclusively and accurately pinpointing human touch and helping humans/robots recognize and response to dangers, such as fire, water, and fracture, have yet to be reported.

(Page 13, Line 223) For a proof of concept, we fabricated an intelligent protective glove for a robot hand, which incorporated SIH fibers (Figure 4a) and integrated a circuit system (Figure S20).

(Page 17, Line 296) Secondly, we realize the perceptual function by developing a surface capacitive sensing system instead of piezoresistive, piezoelectric, or triboelectric principle, which endows our textiles selectively respond only to human touch, not to any external pressure stimulus.

In the supplementary information:

(Page 10, Line 100)

Figure S20 The circuit diagram of the perceptual textiles shown in Figure 4a.

Comment #2:

The biggest problem of ionic hydrogel is dehydration, which affects the overall stability of the device. Although the authors have made long-term storage tests on the conductivity of the SIH fiber, it cannot meet such high humidity conditions (43% humidity) in practical applications. Therefore, it is recommended that the authors supplement the stability of the SIH fiber under actual environmental conditions to evaluate its actual use efficiency.

Reply:

Thank you for raising this point. We agree with you that one of the biggest problems of ionic hydrogel is dehydration which may induce instability of the devices. In this work, we incorporated hydrophilic glycerol and water-soluble ionic liquids into silk hydrogel fibers and obtained ionic fibers with high stability in ambient conditions. The hydrophilicity of glycerol endows it with excellent water retention properties. The water-soluble ionic liquids also reduce the evaporation of water. Silk proteins containing lots of amino acid residues (~30%) with hydrophilic groups also contribute to water retention. These characteristics give our fibers good stability.

According to your comments, we further checked the relative humidity in various regions. We collected the humidity data of the typical cities in China and the world located in different areas, including Haerbin, Beijing, Wuhan, Chengdu, Shanghai, Guangzhou, London (UK), Tokyo (Japan), Canberra (Australia), Ottawa (Canada), New York (US), and Brasilia (Brazil). The monthly average humidity of these cities is shown in Figure S17 and Figure S18. As shown in Figure S17a, from December to May, the humidity in Beijing is around 43% and from June to November, it is higher than 43%. As shown in Figure S17 and S18, all of the monthly average humidity of Haerbin, Wuhan, Chengdu, Shanghai, Guangzhou, London (UK), Tokyo (Japan), Canberra (Australia), Ottawa (Canada), New York (US), and Brasilia (Brazil) are higher than 43%. Based on these data, it can be considered that 43% is a relatively low humidity value in the actual environment.

According to your suggestion, we added related information in the revised manuscript and supplementary information. For your convenience, we copied the related parts in the following:

In the manuscript:

(Page 11, Line 192) We investigated the effect of long-term storage on the ionic conductivity of SIH fibers and found a slight decrease from $0.45 \text{ S} \cdot \text{m}^{-1}$ to $0.43 \text{ S} \cdot \text{m}^{-1}$ after being stored at 43% humidity for 21 days (Figure S16). It is noted that 43% is a relatively low humidity value in actual environment (Figure S17 and S18). The excellent stability of the fiber can be ascribed to existence of water retaining glycerol. Besides, the water-soluble ionic liquids and the silk fibroin that contains lots of amino acid residues (~30%) with hydrophilic groups also reduce the evaporation of water.

In the supplementary information:

Figure S17. The monthly average humidity throughout the year of typical cities of China (2009-2018). (a) Beijing City; (b) Haerbin City; (c) Wuhan City; (d) Chengdu City; (e) Shanghai City; (f) Guangzhou City. The red dashed lines represent the humidity of 43%. (Data source: <http://www.weather.com.cn>)

Figure S18. The monthly average humidity throughout the year of typical cities in the world (2005-2015 for London, 1985-2015 for other cities). (a) London, UK; (b) Tokyo, Japan; (c) Canberra, Australia; (d) Ottawa, Canada; (e) New York, US; (f) Brasilia, Brazil.

Brazil. The red dashed lines represent the humidity of 43%. (Data source: <http://www.timeanddate.com/weather>)

Comment #3:

3. Regarding the comparison with previous tactile textiles, the authors propose four merits. However, I have a different perspective. Firstly, commercial silver plated nylon conductive yarns also have excellent conductivity, excellent mechanical strength and ductility, and relatively low critical costs. Secondly, piezoelectric or triboelectric modes seem to have more advantages as active electrical response mechanisms compared to passive capacitive modes. Thirdly, I don't understand the difference between human contact and external pressure stimulation, they can be regarded as the same. Lastly, conductive connections are still required between SIH fibers of different channels, how to reduce external circuits. The author's final highlight is completely unconvincing.

Reply:

Thank you for sharing your perspective on different tactile textiles. Following is a point-to-point response to your concerns and our viewpoint on the different fibers/textiles.

(1) We agree with you that commercial silver plated nylon conductive yarns have excellent conductivity and excellent mechanical strength. And their ductility is higher than that of metal wires. However, our ionic fibers have significant differences with the silver plated nylon yarns in terms of stretchability and electrical stability. Firstly, our ionic fiber possesses high elongation at break (>350%), compared to 44% for silver-plated nylon yarn (Figure R1b). Secondly, the electrical properties of our fiber is stable under large strains (>350%). It can maintain 70% electric performance before breaking

(Figure 3f). The silver-plated nylon yarn loses its electrical conductivity at less than 50%, showing sudden drops as the strands in the yarn break one by one (Figure R1c). It only retains 7% electric performance when it is nearly completely broken.

Figure R1. (a) An optical picture of a commercial silver plated nylon conductive yarn. (b) Stress-strain curve of the silver plated nylon yarn. (c) Current response of the silver plated nylon yarn to strain.

- (2) The detection of pressure based on the electrical signals generated by piezoelectric and triboelectric materials in response to pressure is also an effective method. However, piezoelectric and triboelectric textiles respond to pressure from most external objects rather than just the touch from human hands, similar to piezoresistive textiles (*Nat. Electron.* 2021, 4, 193-201). Our work constructs a capacitive sensing system, enabling its special and precise pinpointing of human touch.
- (3) In this work, there is a great difference between human contact and external pressure stimulation. When a finger touches the textile, the circuits we developed instantly form a close circuit (Figure 5b and 5d), which recognizes the touch and locates the touch point. Although the touch process also generates pressure on the textile, the pressure is not the reason why our perceptual textile generates electrical responses. The electrical properties (resistance) of our fibers are very stable when they are pressed (Figure 3d).

That is, our textile only responds to finger touch, preventing it from responding to accidental contact (pressure) from other objects. Therefore, based on these characteristics, our textiles can be used as a new type of textile human-machine interface to control other terminals through finger touch. In contrast, the textiles based on the piezoresistive mode in the reference (*Nat. Electron.* 2021, 4, 193-201) are able to generate electrical response to the pressure from all external objects. Therefore, their textiles are not suitable for human-machine interfaces, but rather for pressure sensors.

(4) As can be seen from the Figure 5d, our perceptual textile needs only four external wires to accurately identify the touch and its location. As shown in the Figure R2a, the tactile textile in the literature (*Nat. Electron.* 2021, 4, 193-201) requires lots of wires to be connected to all the warp and weft fibers in the textile. Figure R2b demonstrates their tactile gloves with a large number of wires. Therefore, compared to the piezoresistive tactile textiles, our perceptual textiles greatly reduce the number of external wires.

Figure R2. The circuits of tactile textiles based on piezoresistive modes in Ref (*Nat. Electron.* 2021, 4, 193-201). (a) A modified electrical-grounding-based isolation circuit architecture for passive sensing array readout. (b) Digitally designed and automatically knitted full-sized tactile sensing gloves.

Accordingly, we have revised the related discussion part in the manuscript. For your convenience, we copied the related parts in the following:

In the manuscript:

(Page 17, Line 294) Compared to previous tactile textiles⁴², endowing our textiles with inherent stretchability, flexibility, and electrical stability. a surface capacitive sensing system instead of piezoresistive, piezoelectric, or triboelectric principle, which endows our textiles selectively response only to human touch, not to any external pressure stimulus. Lastly, our perceptual textiles minimize wiring requirements and eliminate the need for complex external circuits.

Comment #4:

4. It is necessary to characterize the distribution form and state of ionic liquids in the SIH fibers through experiments.

Reply:

Thank you for your suggestion on this point. Accordingly, we performed surface analysis and depth profiling of silk fibers through X-ray photoelectron spectroscopy (XPS) tests. As shown in Figure S8, the XPS curves of 3.0 M-S show significant characteristic peaks of elements F 1s and B 1s attributed to the ionic liquid [Emim]BF₄, indicating the successful incorporation of the ionic liquid into silk fibers. Moreover, both the XPS curves recorded on the surface and inside (200 nm depth) of silk fibers (Figure S8) exhibit these characteristic peaks, indicating that the ionic liquid is distributed on the surface and inside of 3.0M-S. Figure S9 demonstrates the results of depth profiling of XPS from the surface to 200 nm depth of 3.0 M-S with a step depth of 20 nm. It can be seen that the peak

intensities of F 1s and B 1s on the surface of 3.0 M-S are lower than those of F 1s and B 1s inside the fiber. Atomic concentration results proved that the contents of F and B atoms in the fiber surface are less than those of F and B atoms in the fiber interior (Figure S9f), indicating that the ionic liquids are relatively less abundant on the surface. For the control group of 0.0 M-S, there are no obvious F 1s and B 1s signals in the XPS curves of surface and depth profiles (Figure S10-S11). These results are consistent with those of energy dispersive spectrometry (EDS) tests. As shown in Figure 2c, the element F is distributed on the surface and inside of 3.0 M-S, while there is no F in the control group 0.0 M-S. Accordingly, we have added these results and analysis into manuscript and supplementary. For your convenience, we copied the related parts and added contents in the following:

In the manuscript:

(Page 7, Line 108)

Figure 2. Composition and structure analysis of different SIH fiber samples. (a-b) TG (a) and DTG (b) curves. (c) SEM images and corresponding EDS mapping. (d) FTIR spectra. (e) Secondary structure contents of different SIH fiber samples. (f) POM images of SIH fibers. Scale bar, 500 μm .

(Page 8, Line 126) X-ray photoelectron spectroscopy (XPS) tests were also performed to analyze the distribution of [Emim]BF₄ in the fibers. The XPS curves recorded from the surface to the inside of 3.0M-S showed characteristic peaks of elements F 1s and B 1s (Figure S8-S9), indicating the successful incorporation of [Emim]BF₄. In contrast, no B 1s and F 1s signals were observed in the XPS curves of 0.0 M-S (Figure S10-S11).

(Page 20, Line 373) Surface analysis and depth profiling of fibers were performed using a XPS instrument (Ulvac-Phi Inc., PHI Quantera II).

In the supplementary information:

(Page 5, Line 42)

Figure S8 XPS analysis of the surface and inside (depth 200 nm) of 3.0 M-S. (a) XPS survey spectra of the fibers. (b-f) High resolution C 1s (b), N 1s (c), O 1s (d), F 1s (e), and B 1s (f) profiles of the fibers.

Figure S9 Depth profiling of XPS for 3.0 M-S. (a-e) XPS profile curves of C 1s (a), N 1s (b), O 1s (c), F 1s (d), and B 1s (e) for 3.0 M-S. (f) Atom concentration of C, N, O, F, and B in 3.0 M-S.

Figure S10 XPS analysis of the surface and inside (depth 200 nm) of 0.0 M-S. (a) XPS survey spectra of the fibers. (b-f) High resolution C 1s (b), N 1s (c), O 1s (d), F 1s (e), and B 1s (f) profiles of the fibers.

Figure S11 Depth profiling of XPS for 0.0 M-S. (a-e) XPS profile curves of C 1s (a), N 1s (b), O 1s (c), F 1s (d), and B 1s (e) for 0.0 M-S.

Comment #5:

5. In Figure 1, it can be seen that ionic liquids are coated on the surface of silk fibroin fibers through infiltration, but there is not a strong interaction between the two. Therefore, how to maintain stability between the two interfaces. Ionic liquids are on the surface and their stability is more difficult to ensure. It is recommended to place ionic liquids on the core layer more appropriately.

Reply:

As discussed in the above and proved by XPS and EDS, the ionic liquid is incorporated in the fiber instead of being coated on the surface. Accordingly, we revised Figure 1a to avoid misunderstanding. For your convenience, we copied the revised figure in the following:

In the manuscript:

(Page 5, Line 78)

Figure 1. Fabrication of silk fibroin-based ionic hydrogel (SIH) fibers. (a) Scheme diagram for the fabrication of SIH fibers. (b) Photograph of a roll of SIH fibers. (c-d) Typical SEM images of SIH fibers, showing the cross-sectional (c) and surface (d) morphology. (e) Photograph of a SIH fiber coil on a color image, showing its transparency and appearance. (f) A plain weave textile made of SIH fibers, demonstrating the weavability of SIH fibers. (g) An electroluminescent “SILK” pattern with SIH fibers serving as the flexible and transparent electrodes.

REVIEWERS' COMMENTS

Reviewer #2 (Remarks to the Author):

[Note from the Editor: Reviewer #2 was asked to look also over the response given to Reviewer #1]

All issues raised by the reviewer have been addressed in the revised manuscript. The revised version is recommended for publication in NC as it is.